# ROBUST AND EFFICIENT COLLABORATIVE LEARNING

## ABSTRACT

Collaborative machine learning is challenged by training-time adversarial behaviors. Existing approaches to tolerate such behaviors either rely on a central server or induce high communication costs. We propose *Robust Pull-based Epidemic Learning (RPEL)*, a novel, scalable collaborative approach to ensure robust learning despite adversaries. RPEL does not rely on any central server and, unlike traditional methods, where communication costs grow in $\mathcal{O}(n^2)$ with the number of nodes $n$, RPEL employs a pull-based epidemic communication strategy that scales in $\mathcal{O}(n \log n)$. By pulling model parameters from small random subsets of nodes, RPEL significantly lowers the number of required messages without compromising convergence guarantees, which hold with high probability. Empirical results demonstrate that RPEL maintains robustness in adversarial settings, competes with all-to-all communication accuracy, and scales efficiently across large networks.

## 1 INTRODUCTION

Collaborative machine learning has recently gained traction as a solution for data-intensive, privacy-sensitive tasks where data resides across multiple nodes (Kairouz et al., 2021; McMahan et al., 2017; Lian et al., 2017; Koloskova et al., 2020). This approach, however, faces significant challenges in the presence of malfunctioning or malicious nodes that can disrupt the learning process by sending corrupted or misleading updates Lamport et al. (2019); Blanchard et al. (2017). Robustness against so-called Byzantine attacks is critical, especially as machine learning applications expand into sensitive domains such as healthcare, autonomous systems, and finance, where incorrect model updates could lead to catastrophic outcomes. Prior approaches that cope with adversarial nodes have relied on some form of robust aggregation to mitigate their impact; however, those approaches generally relied on a trusted centralized server (Blanchard et al., 2017; Yin et al., 2018; Chen et al., 2017; Allen-Zhu et al., 2020; Karimireddy et al., 2020; Allouah et al., 2023).

The decentralized, peer-to-peer communication setting offers a promising framework for collaborative machine learning, eliminating the need for a central coordinator, thereby enhancing scalability (Lian et al., 2017; Koloskova et al., 2019; Vogels et al., 2023; Lian et al., 2018; Lu and Wu, 2020). However, the absence of a trusted server exacerbates the challenge of ensuring robustness. Existing robust peer-to-peer learning algorithms generally depend on fully-connected (or at least dense) networks, with communication complexity that scales quadratically with the number of nodes (Farhadkhani et al., 2023; Gorbunov et al., 2022; El-Mhamdi et al., 2021). While ensuring robustness, these approaches induce high overhead and network congestion, creating bottlenecks in decentralized environments. The question of whether it is possible to devise a scalable, communication-efficient scheme that can uphold robustness across heterogeneous nodes in decentralized peer-to-peer networks remained open.

We contribute to answering this question in the affirmative by presenting *Robust Pull-based Epidemic Learning (RPEL)*, a novel collaborative learning approach that uses a randomized, epidemic-based communication scheme in which each node periodically pulls model updates from a small, random subset of peers. This approach significantly decreases the communication burden (compared to traditional peer-to-peer solutions) while preserving robustness against adversarial nodes. RPEL leverages the convergence properties of epidemic-based protocols, ensuring that information about the model parameters spreads rapidly across nodes with minimal communication overhead.

A key technical difficulty in robust peer-to-peer settings lies in determining the number of attackers each honest node could be exposed to. This information is crucial for designing effective robust aggregation rules, e.g. (Blanchard et al., 2017; Yin et al., 2018). The possibility of the attackers' positions on the communication graph to be chosen in an adversarial manner adds an extra layer of complexity. Consequently, most current works typically make strong assumptions about the graph connectivity and induce a high communication cost (Gaucher et al., 2025; He et al., 2022; Gaucher et al., 2024). We show that such a high connectivity and communication cost is not needed. Essentially, while previous approaches require each node to communicate with $\Theta(n)$ neighbors, our framework overcomes this limitation by communicating only with $\mathcal{O}(\log(n))$ nodes.

**Contributions.** Our contributions can be summarized as follows:

- **Efficient robust learning protocol.** We introduce RPEL, a randomized, epidemic-style distributed learning algorithm, designed to tolerate adversaries while minimizing the need for extensive node-to-node interactions. Additionally, we propose a protocol for selecting the number of sampled neighbors that achieves efficient scalability while ensuring robustness with minimal communication overhead.

- **Theoretical guarantees.** We establish rigorous convergence guarantees of our RPEL protocol considering a general non-convex setting, in the presence of data heterogeneity and under mild assumptions widely adopted in the Byzantine-robust optimization community. Our analysis ensures robustness against any omniscient attack, with the capability of exchanging distinct updates with different honest nodes even during the same iteration. We demonstrate that the robustness properties of RPEL hold with high probability, providing strong guarantees in practical deployments. We do so by introducing the key concept of Effective adversarial fraction, a high-probability upper bound on the number of selected attackers. This concept provides a nuanced understanding of the system's ability to tolerate adversaries compared to robust, fixed-graph decentralized learning approaches.

- **Experiments.** We demonstrate that RPEL works well in practice, providing good results on MNIST and CIFAR-10 datasets with up to $20\%$ adversarial nodes within a decentralized system. Notably, RPEL proves to be highly competitive with state-of-the-art all-to-all robust methods, achieving comparable accuracy with a much cheaper communication budget. Additionally, for the same communication cost, RPEL shows better robustness than the baseline fixed-graph methods. Furthermore, we highlight the scalability advantages of RPEL's randomized communication strategy through a suite of simulations capturing the impact of the number of selected nodes on the Effective adversarial fraction.

## 2 RELATED WORK

**Byzantine distributed learning.** Learning in a distributed setting becomes significantly challenging in the presence of Byzantine adversaries namely nodes that may deliberately send malicious or corrupted updates. In the context of Federated Learning, i.e., when a trusted server is available to orchestrate the learning procedure, the use of robust aggregation rules can effectively mitigate the influence of an adversary (Allouah et al., 2023; Yin et al., 2018; Blanchard et al., 2017).

**Robust Decentralized Learning.** Unlike centralized approaches, Decentralized Learning (DL) allows clients to communicate directly with each other to share updated model versions and converge to a common optimization objective. The most popular DL approach is gossip averaging, in which each client iteratively updates their current model using a weighted combination of their neighbors' models (Boyd et al., 2005; Koloskova et al., 2019; Berthier et al., 2020). Since the gossip operation relies on non-robust averaging, this approach fails in the presence of attacks Blanchard et al. (2017). Some DL approaches were, however, recently proposed to ensure robustness to such attacks. Nearest Neighbor Averaging (NNA) Farhadkhani et al. (2023) was shown to be effective in the all-to-all communications setting. For sparse graphs, client-level clipping, He et al. (2022) ensures that the model update remains within a controllable distance from the current state, thereby limiting the impact of adversaries: each honest node fixes the clipping threshold to $\tau_i^t = \sqrt{\frac{1}{\delta(i)} \sum_{j \in \mathcal{H}} W_{ij} \mathbb{E} \left\| \mathbf{x}_i^{t+1/2} - \mathbf{x}_j^{t+1/2} \right\|^2}$, where $\delta(i) = \sum_{j \in \mathcal{B}} W_{ij}$ and $W$ is the gossip matrix. However, computing this threshold is impossible without knowing the attackers' identity. Although they do propose another threshold choice that

can be implemented in practice, it is not covered by their theoretical analysis. A more practical way to fix the clipping threshold, for sparse communication graphs, was proposed in Gaucher et al. (2025). Inspired by NNA, the proposed algorithm clips the $2b$ largest updates received by each honest node, where $b$ is the maximum total adversarial neighbors' weight an honest node can have. The theoretical guarantees, however, involve the honest subgraph and, therefore, largely depend on the location of adversarial nodes on the graph. Remove-Then-Clip (RTC) Yang and Ghaderi (2024), a similar approach to Gaucher et al. (2025), removes neighbors with the furthest model updates, before clipping the rest of the neighbors, using a clipping threshold that is possible to implement (as opposed to He et al. (2022)). Nevertheless, just like Gaucher et al. (2025), the theoretical guarantees of RTC also depend on the properties of the honest subgraph. Another limitation of these robust, sparse decentralized methods is their reliance on a highly connected communication graph to ensure the connectivity of the subgraph of honest nodes. Specifically, for a graph of $n$ nodes among which $b$ are adversarial, the communication graph must be at least $2b$-connected, which is a significant limitation for large-scale networks.

**Epidemic Learning.** The advantages of randomization in peer-to-peer learning were highlighted by De Vos et al. (2024), which introduced Epidemic Learning as a powerful to address the high communication costs that often hinder the practical deployments of Decentralized Learning (Koloskova et al., 2019). In thus paper, we introduce a Pull variant of Epidemic Learning that differs from the one originally studied by De Vos et al. (2024). We demonstrate that pulling preserves the advantages of Epidemic Learning while ensuring robustness in adversarial settings.

## 3 PROBLEM STATEMENT

### 3.1 SETTING

We consider a decentralized, serverless system comprising $n$ nodes $\mathcal{N} = \{1, 2, \ldots, n\}$. Each node $i$ has access to a local dataset sampled from a distribution $\mathcal{D}_i$, which may differ across nodes, reflecting non-IID data conditions. The nodes collectively aim to minimize a global objective function:

$$\min_{\mathbf{x} \in \mathbb{R}^d} F(\mathbf{x}) = \frac{1}{n} \sum_{i=1}^{n} f_i(\mathbf{x}),$$

where $f_i(\mathbf{x}) = \mathbb{E}_{\xi \sim \mathcal{D}_i}[\ell(\mathbf{x}, \xi)]$ is the local loss function for node $i$, and $\mathbf{x} \in \mathbb{R}^d$ represents the model parameters.

### 3.2 THREAT MODEL

In this setting, up to $b < \lfloor n/2 \rfloor$ nodes may exhibit Byzantine behavior, arbitrarily deviating from the prescribed protocol. Controlled by an omniscient adversary, these nodes can send different, potentially malicious updates to the other nodes. The omniscient nature of these attacks implies the adversaries have full knowledge of the honest nodes' updates, the selected nodes at each iteration, and how they execute aggregation. Honest nodes are those which comply with the prescribed protocol.

We denote by $\mathcal{H} \subset \mathcal{N}$ the set of honest nodes in the system. As truthful information about the adversaries' datasets cannot be guaranteed, a more reasonable objective for the honest clients is to minimize:

$$\min_{\mathbf{x} \in \mathbb{R}^d} F_{\mathcal{H}}(\mathbf{x}) = \frac{1}{|\mathcal{H}|} \sum_{i \in \mathcal{H}} f_i(\mathbf{x}). \tag{1}$$

To formalize the robustness of our algorithm, we use the following notion of Byzantine resilience

**Definition 3.1** (Byzantine Resilience). A decentralized algorithm is said to be $(b, \varepsilon)$-resilient if, despite the presence of $b$ Byzantine clients, each honest client $i \in \mathcal{H}$ computes $\hat{\mathbf{x}}_i$ such that:

$$\mathbb{E}\left[\|\nabla F_{\mathcal{H}}(\hat{\mathbf{x}}_i)\|^2\right] \leq \varepsilon,$$

where the expectation is taken over the randomness of the algorithm.

## 3.3 COMMUNICATION MODEL

Nodes communicate via a peer-to-peer network, using an epidemic-style communication strategy:

- **Synchronous setting**: All the participating nodes possess a global clock indicating the current iteration. This entails that the peer selection process at each iteration is uniform among the whole set of participants.

- **Randomized Communication**: In each communication round, all honest nodes pull model updates from a randomly chosen subset of $s$ peers. This pull-based approach to Epidemic Learning was not explored in the work of De Vos et al. (2024), which focused on the non-robust push-based Epidemic Learning where nodes select recipients for their model updates. The pull-based approach is especially important in Byzantine settings because it prevents attackers from injecting malicious updates to all the honest nodes at each iteration.

## 4 ROBUST EPIDEMIC LEARNING

In this section, we first present the RPEL algorithm, designed to guarantee Byzantine robustness in the randomized peer-to-peer communication setting presented in the previous section. We then introduce the concept of Effective adversarial fraction, a key quantity in RPEL that allows a principled approach to the choice of the algorithm's hyperparameters.

### 4.1 ALGORITHM

Algorithm 1 outlines the general procedure executed by each node. This follows a pull-based Epidemic Learning approach, where each node randomly samples $s$ other nodes and updates its model using the $s + 1$ local models, including its own.

The goal of the aggregation step is to guarantee robustness to the selected adversaries, the number of which is upper bounded by $\hat{b}$ with a high probability. Depending on the number of nodes in the graph, the number of sampled neighbors, and the total number of iterations, this upper bound is intuitively smaller than $b$, the true number of adversarial nodes in the whole graph. We discuss this upper bound in more detail in Section 4.2 and Section 6.1.

To guarantee such robustness, we analyze the use of robust aggregation rules, satisfying $(s, \hat{b}, \kappa)$-robustness, which we introduce in Section 5 (Definition 5.1).

---

**Algorithm 1** RPEL algorithm

---

**Require:** number of iterations $T$, number of neighbors $s$, momentum coefficient $\beta$, communication
  step size $\rho$, effective number of adversaries $\hat{b}$, Aggregation Rule $\mathcal{R}$
1: **for** $t = 1 \ldots T$ **do**
2:      **for** $i \in \mathcal{H}$ in parallel **do**
3:          Randomly sample a data point $\xi_i^t$ from $\mathcal{D}_i$
4:          Compute a local stochastic gradient $g_i^t = \nabla f_i(\mathbf{x}_i^t, \xi_i^t)$
5:          Update local momentum $m_i^t = \beta m_i^{t-1} + (1 - \beta)g_i^t$
6:          Optimization step: $\mathbf{x}_i^{t+1/2} = \mathbf{x}_i^t - \eta m_i^t$
7:          Randomly sample a set $S_i^t$ of $s$ nodes
8:          Receive $\mathbf{x}_j^{t+1/2}$ from each $j \in S_i^t$        ▷ Selected attackers send corrupted updates
9:          Aggregation step: $\mathbf{x}_i^{t+1} = \mathcal{R}\left(\mathbf{x}_i^{t+1/2}; \left\{\mathbf{x}_j^{t+1/2}, j \in S_i^t\right\}\right)$
10:      **end for**
11: **end for**
12: **for** $i \in \mathcal{H}$ **do**
13:      Output $\hat{\mathbf{x}}_i \sim \mathcal{U}\{\mathbf{x}_i^1, \ldots, \mathbf{x}_i^T\}$
14: **end for**

---

## 4.2 Effective adversarial fraction

Randomized communication makes it possible to reason about the adversarial fraction similarly to federated learning. Although the adversaries in the peer-to-peer context can have a much stronger impact, since they can send different updates to different clients, having guarantees on the total adversarial fraction is necessary from a practical point of view.

Through randomization, at each iteration $t$, each honest node $i$ gets a random number $b_i^t$ of attackers. This number is drawn from a hypergeometric distribution $b_i^t \sim \mathrm{HG}(n-1, b, s)$.

For $\hat{b}$ such that $b/n < \hat{b}/s+1 < 1/2$, define the following event

$$\Gamma = \left\{ \forall t \leq T, \forall i \in \mathcal{H}, b_i^t \leq \hat{b} \right\}. \tag{2}$$

The following lemma shows that logarithmic sampling is enough for our algorithm at scale.

**Lemma 4.1.** *If the number of samples $s$ satisfies :*

$$s \geq \left\lceil \max \left\{ \frac{1}{(1/2 - b/n)^2}, \frac{3}{b/n} \right\} \ln \left( \frac{4T|\mathcal{H}|}{1-p} \right) \right\rceil + 2, \tag{3}$$

*then there exits $\hat{b}$ such that $\Gamma$ holds with probability at least $p$ and $\frac{\hat{b}}{s+1} \in \mathcal{O}\left(\frac{b}{n}\right)$*

Lemma 4.1 emphasizes that logarithmic scaling of the number of selected nodes $s$ is sufficient to ensure that the event $\Gamma$ occurs with high probability. Indeed, assuming the number of Byzantine clients grows linearly in $n$ (which is equivalent to fixing the total Byzantine fraction $b/n$), one only needs to increase $s$ logarithmically in $n$. This illustrates the efficiency of the RPEL framework, demonstrating that robustness can be achieved with a communication complexity of $\mathcal{O}(n \log n)$. In the rest of the paper, $\hat{b}$ is set such that with probability at least $p$, each honest node has at most $\hat{b}$ adversaries throughout the whole learning. This is allowed by virtue of Lemma A.4 in the Appendix, which gives a sufficient condition (Equation (7)) on the choice of $s$ and $\hat{b}$ for the event $\Gamma$ to hold with at least probability $p$.

We note $\hat{h} = s + 1 - \hat{b}$, which is a lower bound on the number of honest nodes guaranteed to be selected by each node at each iteration (with probability at least $p$). We call the *Effective adversarial fraction* the ratio $\frac{\hat{b}}{s+1}$.

## 5 Theoretical Analysis

We now present our main theoretical results demonstrating the finite-time convergence of Algorithm 1.

### 5.1 Single iteration reduction

To formalize and quantify the robustness of a given aggregation rule, we use the following notion of $(s, \hat{b}, \kappa)$-robustness. This provides an upper bound on the error between the aggregation output and the honest average with the variance of the input of honest nodes.

**Definition 5.1** ($(s, \hat{b}, \kappa)$-robustness (Allouah et al., 2023))**.** Let $\hat{b} \leq s/2$ and $\kappa \geq 0$. An aggregation rule $\mathcal{R}$ is said to be $(s, \hat{b}, \kappa)$-robust if for any vectors $v_1, \ldots, v_{s+1} \in \mathbb{R}^d$, and any set $\mathcal{U} \subseteq [s+1]$ of size $s + 1 - \hat{b}$,

$$\left\| \mathcal{R}(v_1, \ldots, v_{s+1}) - \overline{v}_{\mathcal{U}} \right\|^2 \leq \frac{\kappa}{|\mathcal{U}|} \sum_{i \in \mathcal{U}} \left\| v_i - \overline{v}_{\mathcal{U}} \right\|^2,$$

with $\overline{v}_{\mathcal{U}} \coloneqq \frac{1}{|\mathcal{U}|} \sum_{i \in \mathcal{U}} v_i$.

The next lemma demonstrates that each step of Algorithm 1, when using a robust aggregation rule, effectively reduces the variance among the honest nodes and quantifies the amount of bias introduced.

**Lemma 5.2.** *Assuming $\mathcal{R}$ satisfies Definition 5.1, and assuming Equation (7) is satisfied, then the iterates of Algorithm 1 with $\mathcal{R}$ satisfy the following reduction at any iteration $t$:*

$$\mathbb{E}_t \left[ \left\| \overline{\mathbf{x}_{\mathcal{H}}^{t+1}} - \overline{\mathbf{x}_{\mathcal{H}}^{t+1/2}} \right\|^2 \right] \leq \left( \kappa + \frac{|\mathcal{H}| - \hat{h}}{(|\mathcal{H}| - 1)|\mathcal{H}|\hat{h}} \right) \frac{1}{|\mathcal{H}|} \sum_{i \in \mathcal{H}} \left\| \mathbf{x}_i^{t+1/2} - \overline{\mathbf{x}_{\mathcal{H}}^{t+1/2}} \right\|^2,$$

$$\mathbb{E}_t \left[ \frac{1}{|\mathcal{H}|} \sum_{i \in \mathcal{H}} \left\| \mathbf{x}_i^{t+1} - \overline{\mathbf{x}_{\mathcal{H}}^{t+1}} \right\|^2 \right] \leq \left( 6\kappa + 6\frac{|\mathcal{H}| - \hat{h}}{(|\mathcal{H}| - 1)\hat{h}} \right) \frac{1}{|\mathcal{H}|} \sum_{i \in \mathcal{H}} \left\| \mathbf{x}_i^{t+1/2} - \overline{\mathbf{x}_{\mathcal{H}}^{t+1/2}} \right\|^2.$$

*where the expectation is taken on the random sampling at iteration $t$.*

As a consequence of this lemma and a classic result described in Section A.3, a way to ensure convergence of RPEL is to provide a robust aggregation rule satisfying Definition 5.1 with $\kappa + 1/\hat{h} < \frac{1}{6}$.

## 5.2 CONVERGENCE RATES

To prove the convergence, we use the following assumptions on the objective functions of the honest clients. The first two are standard in first-order stochastic optimization (Bottou et al., 2018), and the third one is standard in robust optimization (Allouah et al., 2024b; Data and Diggavi, 2021).

**Assumption 5.3.** ($L$-Smoothness). $\forall i \in \mathcal{H}, \forall \mathbf{x}, \mathbf{y} \in \mathbb{R}^d$, we have

$$\|\nabla f_i(\mathbf{x}) - \nabla f_i(\mathbf{y})\| \leq L \|\mathbf{x} - \mathbf{y}\|.$$

**Assumption 5.4.** (Bounded stochastic noise). $\forall i \in \mathcal{H}, \forall \mathbf{x} \in \mathbb{R}^d$, we have

$$\mathbb{E}_{\xi \sim \mathcal{D}_i} \left[ \|\nabla \ell(\mathbf{x}, \xi) - \nabla f_i(\mathbf{x})\|^2 \right] \leq \sigma^2.$$

**Assumption 5.5.** (Bounded heterogeneity). $\forall i \in \mathcal{H}, \forall \mathbf{x} \in \mathbb{R}^d$, we have

$$\frac{1}{|\mathcal{H}|} \sum_{i \in \mathcal{H}} \|\nabla f_i(\mathbf{x}) - \nabla F_{\mathcal{H}}(\mathbf{x})\|^2 \leq G^2. \tag{4}$$

We now present the main theoretical result of our paper, showcasing the convergence of both previous aggregations in the non-convex setting.

**Theorem 5.6** (Convergence of Algorithm 1). *Under the assumptions 5.3, 5.4 and 5.5, assuming that $s$ and $\hat{b}$ satisfy condition 7 and $\mathcal{R}$ satisfies Definition 5.1, for a choice of $\eta$ satisfying Equation (13), for all $i \in \mathcal{H}$, after $T$ iterations, with probability at least $p$ on the sampling procedure*

$$\frac{1}{T} \sum_{t=0}^{T} \mathbb{E}\left[ \|\nabla F_{\mathcal{H}}\left(\mathbf{x}_i^t\right)\|^2 \right] \leq 2\sqrt{\frac{\left( c_2 L\sigma^2 + \frac{432L}{T}\left( \frac{\sigma^2}{n-b} \right) \right) c_0}{T}} + 2\sqrt[3]{\frac{9c_0^2 c_1 n L^2(\sigma^2 + G^2)}{T^2}}$$

$$+ 12\frac{Lc_0}{T} + c_3 G^3,$$

*with $c_0 = 12(F_{\mathcal{H}}(\overline{\theta^0}) - F_{\mathcal{H}}^*), c_1 = \frac{18\alpha(1+\alpha)}{(1-\alpha)^2}, c_2 = 72L\left( \frac{3}{n-b} + 2c_1 + \frac{9\lambda}{2}(2c_1 + 3) \right),$*

*$c_3 = 6\left( 6c_1 + \frac{9\lambda}{2}(4c_1 + 9) \right),$ where $\alpha = 6\kappa + 6\frac{|\mathcal{H}| - \hat{h}}{(|\mathcal{H}| - 1)\hat{h}}$ and $\lambda = \kappa + \frac{|\mathcal{H}| - \hat{h}}{(|\mathcal{H}| - 1)|\mathcal{H}|\hat{h}}.$*

Ignoring the constants and the higher-order terms, we deduce the following corollary.

**Corollary 5.7.** *Under the conditions of Theorem 5.6, when $\mathcal{R}$ satisfies Definition 5.1 with $\kappa = \mathcal{O}\left( \frac{\hat{b}}{s+1} \right)$, with probability at least $p$, Algorithm 1 satisfies $(b, \varepsilon)$-resilience (Definition 3.1) with:*

$$\varepsilon \in \mathcal{O}\left( \sqrt{\frac{\hat{b}+1}{(s+1)T}} \sigma^2 + \frac{\hat{b}+1}{s+1} G^2 \right).$$

**Remark.** Allouah et al. (2023) show that using NNM pre-aggregation followed by standard aggregation rules such as Krum (Blanchard et al., 2017), Geometric Median Chen et al. (2017) or coordinate-wise Median (Yin et al., 2018), it is possible to get $\kappa = \mathcal{O}(b/n)$ when aggregating $n$ vectors, among which $b$ are adversarial. In our setting, these robust aggregation rules would satisfy $\kappa = \mathcal{O}(\hat{b}/s+1)$.

## 5.3 DISCUSSION

The first term in Corollary 5.7 shows an expected convergence rate. Indeed, in the absence of attackers, when taking $s = n - 1$, corresponding to all-to-all communication, we recover the classical SGD rates, i.e. $\mathcal{O}\left(\sqrt{\sigma^2/nT}\right)$ (Ghadimi and Lan, 2013). In the scenario when $s = n - 1$ and $\hat{b} = b$, we get the same vanishing term $\mathcal{O}\left(\sqrt{(b+1)\sigma^2/nT}\right)$ as Farhadkhani et al. (2023), which is the best-known rate even with a trusted server Karimireddy et al. (2020).

Unlike the theoretical guarantees of Allouah et al. (2024a), our non-vanishing term $\mathcal{O}\left(\frac{\hat{b}+1}{s+1}G^2\right)$ does not depend on the stochastic noise $\sigma^2$. Additionally, since by Lemma 4.1 $\frac{\hat{b}}{s+1} \in \mathcal{O}\left(\frac{b}{n}\right)$, this matches the robust federated learning lower bound of $\mathcal{O}\left(\frac{b}{n}G^2\right)$ established by Allouah et al. (2023). This improvement is enabled by our use of variance reduction through local momentum (Polyak, 1964) in Algorithm 1, which was not feasible in the federated learning setup of Allouah et al. (2024a).

# 6 EXPERIMENTS

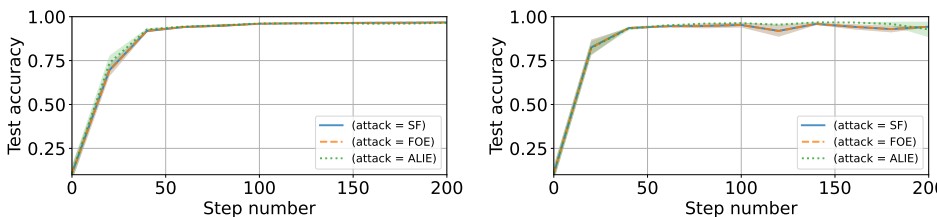

Figure 1: Test accuracies obtained on MNIST. (Left) $n = 100, b = 10, s = 15$ , (right) $n = 30, b = 6, s = 15$.

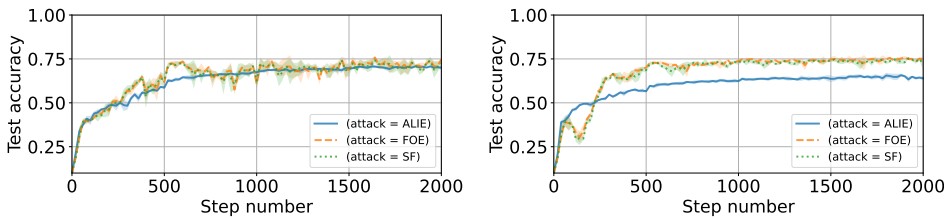

Figure 2: Test accuracies obtained on CIFAR-10 with $n = 20, b = 3$ (Left) $s = 6$, (Right) $s = 19$.

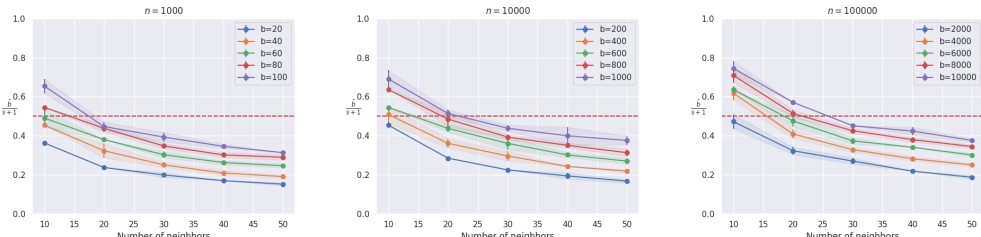

Figure 3: Effective adversarial fraction simulation. For equal adversarial fraction, increasing the number of participants does not require increasing the number of selected neighbors by much.

We empirically evaluate RPEL, assessing its performance across various settings and benchmarking it against state-of-the-art methods. We first describe the experimental setup, and then show some of

results on both MNIST LeCun et al. (2010) and CIFAR-10 Krizhevsky et al. (2014) datasets. All implementation details and the full suite of experiments, including the FEMNIST dataset (Caldas et al., 2019), and the comparison to baselines are included in Section C. For reproducibility, our code is available at `https://anonymous.4open.science/r/RPEL-BF2D/readme`.

## 6.1 EXPERIMENTAL SETUP

**Setting up $s$ and $\hat{b}$**

As stated before, Equation (7) in the Appendix, gives a sufficient condition for $\hat{b}$ to be the maximum number of adversaries selected by each node throughout the whole training. However, this condition can be quite constraining in practice, requiring $s$ to be very large.

As the number of adversarial neighbors of any node $i$ at iteration $t$ follows a Hypergeometric distribution $b_i^t \sim \mathrm{HG}(n-1, b, s)$, it is possible to know the distribution of $\hat{b} = \max_{i \in \mathcal{H}, t \in [1, \dots, T]} (b_i^t)$ for any choice of sampling parameter $s$. Hence it is possible to search for the parameters $s$ and $\hat{b}$ such that $\mathbb{P}\left(\hat{b}/s+1 \leq q\right)$ is very high, whenever that is possible, $q$ being the Effective adversarial fraction the algorithm can support, $1/2$ for instance.

In general, for fixed $n$ and $b$, the larger is $s$ the smaller is the Effective adversarial fraction $\hat{b}/s+1$ (where $\hat{b}$ depends on $s$). A principled way to choose $s$ and $\hat{b}$ is to choose a grid of $s$ values, compute $\hat{b}_s$ for each $s$ by using the confidence interval of a hypergeometric distribution, and then pick the smallest value of $s$ for which $\frac{\hat{b}_s}{s+1}$ is less than $q$. We include a pseudo-code of this method in Section B.

**Attack model and defense**

To make the attacks as strong as possible, we allow the attackers to send distinct malicious updates to the honest clients, even in a single iteration. We also assume the attackers to be omniscient; every time a number $b_i^t$ of adversarial nodes are selected by client $i$, they first look at the honest updates that client $i$ has received at iteration $t$, and depending on the aggregation rule that is used they compute their parameters using one of the state-of-the-art attacks, namely sign flipping (SF) (Li et al., 2020), Fall of Empires (FOE) (Xie et al., 2020) and A Little is Enough (ALIE) (Baruch et al., 2019) and Dissensus (He et al., 2022) for fixed graphs. To defend against these attacks, we use an NNM pre-aggregation rule followed by a coordinate-wise trimmed mean (CWTM) (Yin et al., 2018).

**Heterogeneity**

We use Dirichlet Heterogeneity Hsu et al. (2019) to model the discrepancy between the clients' datasets. A parameter $\alpha$ controls the level of heterogeneity: larger values of $\alpha$ result in more homogeneous datasets among the clients, while smaller values lead to highly diverse data distributions. This model is well suited for capturing the non-IID effects in robust machine learning (Karimireddy et al., 2020; Allouah et al., 2023).

## 6.2 EMPIRICAL RESULTS

**Robustness to high Effective adversarial fraction**

Figure 1 shows the results on MNIST for two settings, in both of which we fix $s = 15$ neighbors to cap the communication cost and set the number of iterations to $T = 200$. First (left in Figure 1), we consider $n = 100$ clients, among which 10 are Byzantine (10%). Using the methodology described in Section 6.1, the Effective Byzantine fraction, in this case, is $0.44$ corresponding to $\hat{b} = 7$. It is worth noting that beyond 10 adversarial clients, in this case, the Effective adversarial fraction exceeds $1/2$, and robust aggregation methods fail. Second (right in Figure 1), we consider $n = 30$ clients among which $b = 6$ are adversarial (20%). The Effective adversarial fraction, in this case, is $0.375$. For both of these settings, RPEL achieves very high accuracy against the three attacks.

Figure 2 (Left) presents the results of RPEL on CIFAR-10 dataset, with $n = 20$ clients among which 3 are adversarial (15%) for $T = 2000$ iterations. We limit the number of selected neighbors to $s = 6$. As $n$ is small in this case, there is a very high probability that at least one of the honest clients will get all three attackers in one of the training rounds. It follows that $\hat{b} = 3$, and the Effective adversarial

fraction is $0.43$. The results show that RPEL allows the clients to achieve close to $75\%$ accuracy on the CIFAR-10 dataset, against the three attacks, despite the Effective adversarial fraction being large.

**Competitive Performance with all-to-all robust algorithms**

To compare RPEL with all-to-all robust Decentralized Learning methods, we conducted additional experiments on the CIFAR-10 dataset. We kept the same setting as before and set $s = 19$ (Figure 2, Right) to allow each honest node to communicate with all the participating nodes at each iteration. Interestingly, the results show that reducing the selection to only $s = 6$ allows us to reach just as good accuracy for a much reduced communication cost. A similar conclusion can be drawn from the MNIST dataset Figure 1, since selecting $s = 15$ out of the $100$ participating nodes is sufficient to maximize the test accuracy. These experimental findings show the practical benefits of randomization, which alleviates the requirement for all-to-all communication to achieve good accuracy.

**Competitive performance with fixed-graph robust algorithms**

For a fair comparison with other robust peer-to-peer methods with fixed graphs, we generate random connected graphs with the same number of communication edges as in our method. We give more details in Section C. We evaluate both the average and worst-case performance. The results, presented in Figure 4 and Figure 5 in Section C show that for the same communication budget, RPEL is much more robust, especially when the communication graph is sparse (low values of $s$). Thanks to the uniform sampling procedure of RPEL, the performance for the worst client is consistently better than the baselines. This suggests that our method also grants better fairness as a byproduct. It is essential to note, however, that fixed-graph methods are particularly suited for inherently sparse settings, where direct connections between all nodes are not feasible. In such cases, the graph topology is given as an input. In contrast, our setting assumes all-to-all communications are available with a uniform cost.

## 6.3 SCALABILITY

Figure 3 presents an empirical simulation of the Effective adversarial fraction for different scenarios of $n$, $b$, and $s$. For each scenario, we use the method described in Section 6.1 to generate 5 independent simulations and construct confidence intervals.

The simulations in Figure 3 highlight the critical role of randomization in large-scale scenarios involving adversaries. As the number of nodes $n$ grows, it is remarkable that the number of sampled neighbors $s$ does not need to increase proportionally to maintain robustness against a fixed proportion of attackers. This scalability is illustrated in the rightmost plot in Figure 3, where even in a network of $n = 100,000$ nodes with an adversarial fraction of $10\%$ (10,000 corrupted nodes), sampling only 30 random neighbors per node is sufficient to maintain an honest majority throughout the entire training process ($T = 200$) for all the $80,000$ honest clients. This is far more efficient than having to ensure the neighborhood of each honest node contains at least 20,001 nodes.

This underscores the power of randomization in mitigating adversarial threats while preserving communication efficiency in decentralized learning. However, conducting machine learning experiments at this scale with tens of thousands of nodes remains computationally prohibitive and is therefore beyond the scope of this paper. Nonetheless, our simulations and theoretical insights provide a strong foundation for future research into scalable and secure decentralized learning in adversarial settings.

## 7 CONCLUSION

This paper presents Robust Pull-based Epidemic Learning (RPEL), an efficient solution to the long-standing problem of learning despite an adversary controlling several nodes in a serverless setting. We establish rigorous convergence guarantees of RPEL, considering a general non-convex setting, in the presence of data heterogeneity and under standard assumptions. Our analysis proves robustness against powerful, omniscient attacks. A key concept underlying our approach is the notion of *Effective adversarial fraction*, a high-probability upper bound on the selected attackers, providing a nuanced understanding of the system's ability to tolerate adversaries. We empirically evaluate RPEL on MNIST and CIFAR-10 datasets with up to $20\%$ adversarial nodes, and we show that RPEL achieves comparable accuracy with all-to-all robust methods for a much cheaper communication budget. Possible future directions include devising more sophisticated sampling techniques and deploying and testing RPEL on large-scale concrete applications.

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

## A  PROOFS

### A.1  NOTATIONS

This subsection presents the different notations we will use throughout the proofs, some of which have already been introduced in the main text.

Recall that in our algorithm, at each iteration $t$, each honest node $i$ samples a set $S_i^t$ among the $n$ nodes present in the graph. For the sake of simplicity of the proof, we consider $|S_i^t| = s + 1$, and we don't enforce that each node selects itself.

Among all the participating nodes, $b$ are supposed to be adversarial. As stated in the paper, we assume with high probability that at most $\hat{b}$ among the selected nodes are adversarial. We note $\hat{h} = s + 1 - \hat{b}$. $\hat{h}$ is a lower bound on the number of honest clients guaranteed to be sampled at each iteration.

In the case of FL-style robust aggregation, the aggregation rule is noted $\mathcal{A}$ and it satisfies $(s, \hat{b}, \kappa)$-robustness Definition 5.1.

The set of honest clients is denoted $\mathcal{H}$. At each iteration $t$, for each $i \in \mathcal{H}$, $\hat{\mathcal{H}}_i^t$ is used to denote a random sample of $\hat{h}$ honest clients among the sampled clients $S_i^t$.

We note $f_i(x) = \mathbb{E}_{\xi \sim \mathcal{D}_i}[\ell(x, \xi)]$ and $F_{\mathcal{H}}(x) = \frac{1}{|\mathcal{H}|} \sum_{i \in \mathcal{H}} f_i(x)$.

At each iteration $t$, each node $i$ computes one stochastic gradient $g_i^t = \nabla \ell(\mathbf{x}_i^t, \xi_i^t)$

$$\mathbf{x}_i^{t+1/2} = \mathbf{x}_i^t - \eta g_i^t. \tag{5}$$

Additionally, for $i, j \in \mathcal{H}$, we use the notation $\mathcal{I}(j \in \hat{\mathcal{H}}_i^t)$ for the indicator variable of the event $\{j \in \hat{\mathcal{H}}_i^t\}$.

Finally, we use the notation $\mathbb{E}_t$ to denote the conditional expectation on all the past events up till iteration $t$, including the random stochastic gradients computed at iteration $t$. In other words, $\mathbb{E}_t$ is the expectation with respect to the random sampling at iteration $t$.

### A.2  USEFUL PROOFS

**Lemma A.1.** *For any honest node $i$, any variables $y_1, \ldots, y_{|\mathcal{H}|}$ and any iteration $t$*

$$\mathbb{E}\left[\frac{1}{\hat{h}} \sum_{j \in \hat{\mathcal{H}}_i^t} y_j\right] = \frac{1}{|\mathcal{H}|} \sum_{j \in \mathcal{H}} y_j,$$

*where the expectation is taken over the randomness of the sampling procedure of $S_i^t$ and $\hat{\mathcal{H}}_i^t$.*

*Proof.* Taking a random sample $S_i^t$ uniformly out of all the participating nodes, and then taking a random number of $\hat{h}$ honest nodes uniformly among the selected nodes $S_i^t$ is equivalent to choosing a random subset of size $\hat{h}$ uniformly out of all the honest participating nodes.

As a consequence of this, we can write:

$$\mathbb{E}\left[\mathcal{I}(j \in \hat{\mathcal{H}}_i^t)\right] = \frac{\hat{h}}{|\mathcal{H}|}.$$

Therefore,

$$\mathbb{E}\left[\frac{1}{\hat{h}} \sum_{j \in \hat{\mathcal{H}}_i^t} y_j\right] = \mathbb{E}\left[\frac{1}{\hat{h}} \sum_{j \in \mathcal{H}} \mathcal{I}(j \in \hat{\mathcal{H}}_i^t) y_j\right]$$

$$= \frac{1}{|\mathcal{H}|} \sum_{j \in \mathcal{H}} y_j.$$

$\square$

**Lemma A.2.** *For $i, j, k \in \mathcal{H}$, such that $j \neq k$ we can compute*

$$\mathrm{Var}(\mathcal{I}(j \in \hat{\mathcal{H}}_i^t)) = \frac{\hat{h}}{|\mathcal{H}|}\left(1 - \frac{\hat{h}}{|\mathcal{H}|}\right),$$

*and*

$$\mathrm{Cov}\left(\mathcal{I}(j \in \hat{\mathcal{H}}_i^t), \mathcal{I}(k \in \hat{\mathcal{H}}_i^t)\right) = \frac{\hat{h}(\hat{h}-1)}{|\mathcal{H}|(|\mathcal{H}|-1)} - \frac{\hat{h}^2}{|\mathcal{H}|^2}.$$

*Proof.* For any $i, j \in \mathcal{H}$, $\mathcal{I}(j \in \hat{\mathcal{H}}_i^t)$ is a Bernoulli variable with the parameter $\frac{\hat{h}}{|\mathcal{H}|}$. This proves the first equality.

If $j \neq k$,

$$\mathrm{Cov}\left(\mathcal{I}(j \in \hat{\mathcal{H}}_i^t), \mathcal{I}(k \in \hat{\mathcal{H}}_i^t)\right) = \mathbb{E}\left[\mathcal{I}(j \in \hat{\mathcal{H}}_i^t)\mathcal{I}(k \in \hat{\mathcal{H}}_i^t)\right] - \mathbb{E}\left[\mathcal{I}(j \in \hat{\mathcal{H}}_i^t)\right]\mathbb{E}\left[\mathcal{I}(k \in \hat{\mathcal{H}}_i^t)\right]$$

$$= \frac{\hat{h}(\hat{h}-1)}{|\mathcal{H}|(|\mathcal{H}|-1)} - \frac{\hat{h}^2}{|\mathcal{H}|^2},$$

which concludes the lemma. $\square$

### A.3 $(\alpha, \lambda)$-REDUCTION

We now present the so-called $(\alpha, \lambda)$-reduction, introduced in Farhadkhani et al. (2023), which is a sufficient condition for convergence in the Byzantine peer-to-peer setting. For clarity, we restate it here.

**Definition A.3** ($(\alpha, \lambda)$-reduction). An algorithm $\mathcal{F}$ is said to satisfy $(\alpha, \lambda)$-reduction if for the inputs $(x_1, \ldots, x_{|\mathcal{H}|})$, the outputs $(y_1, \ldots, y_{|\mathcal{H}|}) = \mathcal{F}(x_1, \ldots, x_{|\mathcal{H}|})$ satisfy:

$$\frac{1}{|\mathcal{H}|}\sum_{i=1}^{|\mathcal{H}|}\|y_i - \overline{y_{\mathcal{H}}}\|^2 \leq \alpha \frac{1}{|\mathcal{H}|}\sum_{i=1}^{|\mathcal{H}|}\|x_i - \overline{x_{\mathcal{H}}}\|^2,$$

$$\|\overline{y_{\mathcal{H}}} - \overline{x_{\mathcal{H}}}\|^2 \leq \lambda \frac{1}{|\mathcal{H}|}\sum_{i=1}^{|\mathcal{H}|}\|x_i - \overline{x_{\mathcal{H}}}\|^2,$$

where $\overline{x_{\mathcal{H}}} = \frac{1}{|\mathcal{H}|}\sum_{i \in \mathcal{H}} x_i$ and $\overline{y_{\mathcal{H}}} = \frac{1}{|\mathcal{H}|}\sum_{i \in \mathcal{H}} y_i$.

As proven in Farhadkhani et al. (2023), when this condition is satisfied by the iterates of any decentralized algorithm, it is sufficient to have $\alpha < 1$ and $\lambda < +\infty$ for the algorithm to satisfy $(b, \varepsilon)$-Byzantine resilience, for some value of $\varepsilon$.

### A.4 THEORETICAL SAMPLING THRESHOLD

**Lemma A.4.** *Recall the event*

$$\Gamma = \left\{\forall t \leq T, \forall i \in \mathcal{H}, b_i^t \leq \hat{b}\right\}. \tag{6}$$

*Let $p < 1$ and assume $s$ and $\hat{b}$ are such that $b/n < \hat{b}/s+1 < 1/2$ and*

$$s \geq \min\left\{n-1, D\left(\frac{\hat{b}}{s}, \frac{b}{n-1}\right)^{-1}\ln\left(\frac{T|\mathcal{H}|}{1-p}\right)\right\}, \tag{7}$$

*with $D(\alpha, \beta) = \alpha \ln(\alpha/\beta) + (1-\alpha)\ln(1-\alpha/1-\beta)$.*

*Then the event $\Gamma$ holds with probability at least $p$.*

## A.5 Proof of Lemma 5.2

**Lemma 5.2.** *Assuming $\mathcal{R}$ satisfies Definition 5.1, and assuming Equation (7) is satisfied, then the iterates of Algorithm 1 with $\mathcal{R}$ satisfy the following reduction at any iteration $t$:*

$$\mathbb{E}_t\left[\left\|\overline{\mathbf{x}_{\mathcal{H}}^{t+1}} - \overline{\mathbf{x}_{\mathcal{H}}^{t+1/2}}\right\|^2\right] \leq \left(\kappa + \frac{|\mathcal{H}| - \hat{h}}{(|\mathcal{H}| - 1)|\mathcal{H}|\hat{h}}\right)\frac{1}{|\mathcal{H}|}\sum_{i \in \mathcal{H}}\left\|\mathbf{x}_i^{t+1/2} - \overline{\mathbf{x}_{\mathcal{H}}^{t+1/2}}\right\|^2,$$

$$\mathbb{E}_t\left[\frac{1}{|\mathcal{H}|}\sum_{i \in \mathcal{H}}\left\|\mathbf{x}_i^{t+1} - \overline{\mathbf{x}_{\mathcal{H}}^{t+1}}\right\|^2\right] \leq \left(6\kappa + 6\frac{|\mathcal{H}| - \hat{h}}{(|\mathcal{H}| - 1)\hat{h}}\right)\frac{1}{|\mathcal{H}|}\sum_{i \in \mathcal{H}}\left\|\mathbf{x}_i^{t+1/2} - \overline{\mathbf{x}_{\mathcal{H}}^{t+1/2}}\right\|^2.$$

*where the expectation is taken on the random sampling at iteration $t$.*

*Proof.* • **First inequality**

For ease of notation, we omit the exponent $t + 1/2$, denoting $\mathbf{x}_i^{t+1/2}$ by simply $\mathbf{x}_i$.

Using the definition of the variance, we write the first inequality as follows.

$$\mathbb{E}_t\left[\left\|\overline{\mathbf{x}_{\mathcal{H}}^{t+1}} - \overline{\mathbf{x}_{\mathcal{H}}}\right\|^2\right] \leq \left(\kappa + \frac{|\mathcal{H}| - \hat{h}}{(|\mathcal{H}| - 1)|\mathcal{H}|\hat{h}}\right)\frac{1}{2|\mathcal{H}|^2}\sum_{i,j \in \mathcal{H}}\|\mathbf{x}_i - \mathbf{x}_j\|^2. \tag{8}$$

Using the aggregation formula of Algorithm 1, we have

$$\overline{\mathbf{x}_{\mathcal{H}}^{t+1}} - \overline{\mathbf{x}_{\mathcal{H}}} = \underbrace{\frac{1}{|\mathcal{H}|}\sum_{i \in \mathcal{H}}\left(\mathcal{A}\left(\mathbf{x}_j, j \in S_i^t\right) - \frac{1}{\hat{h}}\sum_{j \in \hat{\mathcal{H}}_i^t}\mathbf{x}_j\right)}_{T_1} + \underbrace{\frac{1}{|\mathcal{H}|\hat{h}}\sum_{i \in \mathcal{H}}\sum_{j \in \hat{\mathcal{H}}_i^t}\mathbf{x}_j - \frac{1}{|\mathcal{H}|}\sum_{i \in \mathcal{H}}\mathbf{x}_j}_{T_2}.$$

For $T_1$, by the AM-QM inequality, we have

$$\mathbb{E}_t[\|T_1\|^2] \leq \frac{1}{|\mathcal{H}|}\sum_{i \in \mathcal{H}}\mathbb{E}_t\left\|\mathcal{A}\left(\mathbf{x}_j, j \in S_i^t\right) - \frac{1}{\hat{h}}\sum_{j \in \hat{\mathcal{H}}_i^t}\mathbf{x}_j\right\|^2. \tag{9}$$

Since $\mathcal{A}$ satisfies $(s, \hat{b}, \kappa)$-robustness, we have for each $i \in \mathcal{H}$

$$\left\|\mathcal{A}\left(\mathbf{x}_j, j \in S_i^t\right) - \frac{1}{\hat{h}}\sum_{j \in \hat{\mathcal{H}}_i^t}\mathbf{x}_j\right\|^2 \leq \frac{\kappa}{\hat{h}}\sum_{j \in \hat{\mathcal{H}}_i^t}\left\|\mathbf{x}_j - \overline{\mathbf{x}_{\hat{\mathcal{H}}_i^t}}\right\|^2$$

$$= \frac{\kappa}{2\hat{h}^2}\sum_{j,k \in \hat{\mathcal{H}}_i^t}\|\mathbf{x}_j - \mathbf{x}_k\|^2.$$

Taking the expectation on the random sampling of each honest node, and using Lemma A.1, we have:

$$\mathbb{E}_t\left[\left\|\mathcal{A}\left(\mathbf{x}_j, j \in S_i^t\right) - \frac{1}{\hat{h}}\sum_{j \in \hat{\mathcal{H}}_i^t}\mathbf{x}_j\right\|^2\right] \leq \frac{\kappa}{2|\mathcal{H}|^2}\sum_{j,k \in \mathcal{H}}\|\mathbf{x}_j - \mathbf{x}_k\|^2. \tag{10}$$

Plugging Equation (10) in Equation (9), we get

$$\mathbb{E}_t[\|T_1\|^2] \le \frac{\kappa}{2|\mathcal{H}|^2} \sum_{i,j\in\mathcal{H}} \|\mathbf{x}_i - \mathbf{x}_j\|^2.$$

For $T_2$, noticing that $\mathbb{E}_t[T_2] = 0$, thanks to Lemma A.1, and using the independence of $\hat{\mathcal{H}}_i^t$'s, which is a key feature of pull-based Epidemic Learning, we can write :

$$\mathbb{E}_t[\|T_2\|^2] = \mathrm{Var}_t\left(\frac{1}{|\mathcal{H}|\hat{h}} \sum_{i\in\mathcal{H}} \sum_{j\in\hat{\mathcal{H}}_i^t} \mathbf{x}_j\right)$$

$$= \frac{1}{|\mathcal{H}|}\mathrm{Var}_t\left(\frac{1}{\hat{h}} \sum_{j\in\hat{\mathcal{H}}_i^t} \mathbf{x}_j\right)$$

$$= \frac{1}{|\mathcal{H}|\hat{h}^2}\mathrm{Var}_t\left(\sum_{j\in\mathcal{H}} \mathcal{I}(j\in\hat{\mathcal{H}}_i^t)\mathbf{x}_j\right)$$

$$= \frac{1}{|\mathcal{H}|\hat{h}^2}\left(\sum_{j\in\mathcal{H}} \mathrm{Var}_t(\mathcal{I}(j\in\hat{\mathcal{H}}_i^t))\|\mathbf{x}_j\|^2 + \sum_{j\neq k} \mathrm{Cov}_t\left(\mathcal{I}(j\in\hat{\mathcal{H}}_i^t), \mathcal{I}(k\in\hat{\mathcal{H}}_i^t)\right)\mathbf{x}_j^T\mathbf{x}_k\right).$$

Now using Lemma A.2, it follows that:

$$\mathbb{E}_t[\|T_2\|^2] = \frac{1}{|\mathcal{H}|\hat{h}^2}\left(\frac{\hat{h}}{|\mathcal{H}|}\left(1 - \frac{\hat{h}}{|\mathcal{H}|}\right)\sum_{j\in\mathcal{H}}\|\mathbf{x}_j\|^2 + \left(\frac{\hat{h}(\hat{h}-1)}{|\mathcal{H}|(|\mathcal{H}|-1)} - \frac{\hat{h}^2}{|\mathcal{H}|^2}\right)\sum_{j\neq k}\mathbf{x}_j^T\mathbf{x}_k\right)$$

$$= \frac{|\mathcal{H}|-\hat{h}}{|\mathcal{H}|^2\hat{h}}\left(\frac{1}{|\mathcal{H}|}\sum_j\|\mathbf{x}_j\|^2 + \left(\frac{\hat{h}-1}{|\mathcal{H}|-1} - \frac{\hat{h}}{|\mathcal{H}|}\right)\sum_{j\neq k}\mathbf{x}_j^T\mathbf{x}_k\right)$$

$$= \frac{|\mathcal{H}|-\hat{h}}{(|\mathcal{H}|-1)|\mathcal{H}|\hat{h}}\left(\frac{1}{|\mathcal{H}|}\sum_j\|\mathbf{x}_j\|^2 - \left\|\frac{1}{|\mathcal{H}|}\sum_j\mathbf{x}_j\right\|^2\right)$$

$$= \frac{|\mathcal{H}|-\hat{h}}{(|\mathcal{H}|-1)|\mathcal{H}|\hat{h}}\left(\frac{1}{2|\mathcal{H}|^2}\sum_{i,j\in\mathcal{H}}\|\mathbf{x}_i - \mathbf{x}_j\|^2\right),$$

$$\tag{11}$$

which completes the first part of the proof.

- **Second inequality.**

  We write the second inequality as follows:

  $$\frac{1}{2|\mathcal{H}|^2}\sum_{i,j\in\mathcal{H}}\mathbb{E}_t\left[\|\mathbf{x}_i^{t+1} - \mathbf{x}_j^{t+1}\|^2\right] \le \left(6\kappa + 6\frac{|\mathcal{H}|-\hat{h}}{(|\mathcal{H}|-1)\hat{h}}\right)\left(\frac{1}{2|\mathcal{H}|^2}\sum_{i,j\in\mathcal{H}}\|\mathbf{x}_i - \mathbf{x}_j\|^2\right).$$

  $$\tag{12}$$

  For $i, j \in \mathcal{H}$, we have

$$\mathbf{x}_i^{t+1} - \mathbf{x}_j^{t+1} = \left( \mathcal{A}\left(\mathbf{x}_k, k \in S_i^t\right) - \frac{1}{\hat{h}} \sum_{k \in S_i^t} \mathbf{x}_k \right) - \left( \mathcal{A}\left(\mathbf{x}_l, l \in S_j^t\right) - \frac{1}{\hat{h}} \sum_{l \in S_j^t} \mathbf{x}_l \right)$$
$$+ \frac{1}{\hat{h}} \sum_{k \in S_i^t} \mathbf{x}_k - \frac{1}{\hat{h}} \sum_{l \in S_j^t} \mathbf{x}_l.$$

Taking the norm and using the inequality $\|a + b + c\|^2 \leq 3\|a\|^2 + 3\|b\|^2 + 3\|c\|^2$ (which is also an AM-QM inequality), we have

$$\|\mathbf{x}_i^{t+1} - \mathbf{x}_j^{t+1}\|^2 \leq 3 \left\| \mathcal{A}\left(\mathbf{x}_k, k \in S_i^t\right) - \frac{1}{\hat{h}} \sum_{k \in S_i^t} \mathbf{x}_k \right\|^2 + 3 \left\| \mathcal{A}\left(\mathbf{x}_l, l \in S_j^t\right) - \frac{1}{\hat{h}} \sum_{l \in S_j^t} \mathbf{x}_l \right\|^2$$
$$+ 3 \left\| \frac{1}{\hat{h}} \sum_{k \in S_i^t} \mathbf{x}_k - \frac{1}{\hat{h}} \sum_{l \in S_j^t} \mathbf{x}_l \right\|^2.$$

Taking the expectation, summing over all the pairs of honest nodes, and using $(s, \hat{b}, \kappa)$-robustness along with Lemma A.1, we get

$$\frac{1}{2|\mathcal{H}|^2} \sum_{i,j \in \mathcal{H}} \mathbb{E}_t \left[ \|\mathbf{x}_i^{t+1} - \mathbf{x}_j^{t+1}\|^2 \right] \leq 6\kappa \frac{1}{2|\mathcal{H}|^2} \sum_{i,j \in \mathcal{H}} \|\mathbf{x}_i - \mathbf{x}_j\|^2$$
$$+ \frac{3}{2|\mathcal{H}|^2} \sum_{i,j \in \mathcal{H}} \mathbb{E}_t \left[ \left\| \frac{1}{\hat{h}} \sum_{k \in \hat{S}_i^t} \mathbf{x}_k - \frac{1}{\hat{h}} \sum_{l \in \hat{S}_j^t} \mathbf{x}_l \right\|^2 \right].$$

Using the fact that $S_i^t$ and $S_j^t$ are independent, and using Equation (11) which shows

$$\mathrm{Var}\left( \frac{1}{\hat{h}} \sum_{j \in \hat{\mathcal{H}}_i^t} \mathbf{x}_j \right) = \frac{|\mathcal{H}| - \hat{h}}{(|\mathcal{H}| - 1)\hat{h}} \left( \frac{1}{2|\mathcal{H}|^2} \sum_{i,j \in \mathcal{H}} \|\mathbf{x}_i - \mathbf{x}_j\|^2 \right),$$

we get

$$\frac{1}{2|\mathcal{H}|^2} \sum_{i,j \in \mathcal{H}} \mathbb{E}_t \left[ \|\mathbf{x}_i^{t+1} - \mathbf{x}_j^{t+1}\|^2 \right] \leq 6\kappa \frac{1}{2|\mathcal{H}|^2} \sum_{i,j \in \mathcal{H}} \|\mathbf{x}_i - \mathbf{x}_j\|^2$$
$$+ 6 \frac{|\mathcal{H}| - \hat{h}}{(|\mathcal{H}| - 1)\hat{h}} \left( \frac{1}{2|\mathcal{H}|^2} \sum_{i,j \in \mathcal{H}} \|\mathbf{x}_i - \mathbf{x}_j\|^2 \right)$$
$$\leq \left( 6\kappa + 6 \frac{|\mathcal{H}| - \hat{h}}{(|\mathcal{H}| - 1)\hat{h}} \right) \left( \frac{1}{2|\mathcal{H}|^2} \sum_{i,j \in \mathcal{H}} \|\mathbf{x}_i - \mathbf{x}_j\|^2 \right).$$

This concludes the proof of Lemma 5.2.

$\square$

### A.6 PROOF OF THEOREM 5.6

**Theorem 5.6** (Convergence of Algorithm 1). *Under the assumptions 5.3, 5.4 and 5.5, assuming that $s$ and $\hat{b}$ satisfy condition 7 and $\mathcal{R}$ satisfies Definition 5.1, for a choice of $\eta$ satisfying Equation (13), for all $i \in \mathcal{H}$, after $T$ iterations, with probability at least $p$ on the sampling procedure*

$$\frac{1}{T}\sum_{t=0}^{T}\mathbb{E}\left[\|\nabla F_{\mathcal{H}}\left(\mathbf{x}_i^t\right)\|^2\right] \leq 2\sqrt{\frac{\left(c_2 L\sigma^2 + \frac{432L}{T}\left(\frac{\sigma^2}{n-b}\right)\right)c_0}{T}} + 2\sqrt[3]{\frac{9c_0^2 c_1 nL^2(\sigma^2+G^2)}{T^2}}$$
$$+ 12\frac{Lc_0}{T} + c_3 G^3,$$

*with $c_0 = 12(F_{\mathcal{H}}(\overline{\theta^0}) - F_{\mathcal{H}}^*), c_1 = \frac{18\alpha(1+\alpha)}{(1-\alpha)^2}, c_2 = 72L\left(\frac{3}{n-b} + 2c_1 + \frac{9\lambda}{2}(2c_1+3)\right),$*

$c_3 = 6\left(6c_1 + \frac{9\lambda}{2}(4c_1+9)\right),$ *where $\alpha = 6\kappa + 6\frac{|\mathcal{H}|-\hat{h}}{(|\mathcal{H}|-1)\hat{h}}$ and $\lambda = \kappa + \frac{|\mathcal{H}|-\hat{h}}{(|\mathcal{H}|-1)|\mathcal{H}|\hat{h}}$.*

*Proof.* Based on the proof of Theorem 1 from Farhadkhani et al. (2023), we have after $T$ iterations satisfying $\alpha, \lambda$ reduction, if $\eta \leq \frac{1}{12L}$:

$$\frac{1}{T}\sum_{t=0}^{T}\mathbb{E}\left[\|\nabla F_{\mathcal{H}}\left(\mathbf{x}_i^t\right)\|^2\right] \leq \frac{c_0}{\eta T} + c_2 \eta L\sigma^2 + \frac{432\eta L}{T}\left(\frac{\sigma^2}{n-b}\right) + c_3 G^2 + 9c_1 n\eta^2 L^2(\sigma^2+G^2),$$

with $c_0 = 12(F_{\mathcal{H}}(\overline{\theta^0}) - F_{\mathcal{H}}^*)$, $c_1 = \frac{18\alpha(1+\alpha)}{(1-\alpha)^2}$, $c_2 = 72L\left(\frac{3}{n-b} + 2c_1 + + \frac{9\lambda}{2}(2c_1+3)\right)$, $c_3 = 6\left(6c_1 + \frac{9\lambda}{2}(4c_1+9)\right)$.

Taking

$$\eta \leq \min\left\{\frac{1}{12L}, \sqrt[3]{\frac{c_0}{9Tc_1 nL^2(\sigma^2+G^2)}}, \sqrt{\frac{c_0}{T\left(c_2 L\sigma^2 + \frac{432L}{T}\left(\frac{\sigma^2}{n-b}\right)\right)}}\right\}, \quad (13)$$

we get

$$\frac{1}{T}\sum_{t=0}^{T}\mathbb{E}\left[\|\nabla F_{\mathcal{H}}\left(\mathbf{x}_i^t\right)\|^2\right] \leq 2\sqrt[3]{\frac{9c_0^2 c_1 nL^2(\sigma^2+G^2)}{T^2}} + 2\sqrt{\frac{\left(c_2 L\sigma^2 + \frac{432L}{T}\left(\frac{\sigma^2}{n-b}\right)\right)c_0}{T}}$$
$$+ 12\frac{Lc_0}{T} + c_3 G^3$$

$\square$

### A.7 PROOF OF COROLLARY 5.7

**Corollary 5.7.** *Under the conditions of Theorem 5.6, when $\mathcal{R}$ satisfies Definition 5.1 with $\kappa = \mathcal{O}\left(\frac{\hat{b}}{s+1}\right)$, with probability at least $p$, Algorithm 1 satisfies $(b, \varepsilon)$-resilience (Definition 3.1) with:*

$$\varepsilon \in \mathcal{O}\left(\sqrt{\frac{\hat{b}+1}{(s+1)T}}\sigma^2 + \frac{\hat{b}+1}{s+1}G^2\right).$$

*Proof.* In a similar fashion to the proof of Corollary 1 in Farhadkhani et al. (2023), we have:

$$c_1 = \frac{18\alpha(1+\alpha)}{(1-\alpha)^2} \in \mathcal{O}(\alpha),$$

$$c_2 = 72L\left(\frac{3}{n-b} + 2c_1 + \frac{9\lambda}{2}(2c_1+3)\right) = \mathcal{O}(\frac{1}{n} + \alpha + \lambda),$$

$$c_3 = 6\left(6c_1 + \frac{9\lambda}{2}(4c_1+9)\right) = \mathcal{O}(\alpha + \lambda).$$

Since $\alpha = 6\kappa + 6\frac{|\mathcal{H}|-\hat{h}}{(|\mathcal{H}|-1)\hat{h}}$ and $\lambda = \kappa + \frac{|\mathcal{H}|-\hat{h}}{(|\mathcal{H}|-1)|\mathcal{H}|\hat{h}}$. Following Allouah et al. (2023), using NNM pre-aggregation followed by trimmed mean allows to have $\kappa \in \mathcal{O}(\frac{\hat{b}}{s+1})$.

It follows that in both cases, we have:

$$\alpha \in \mathcal{O}\left(\frac{\hat{b}+1}{s}\right), \lambda \in \mathcal{O}\left(\frac{\hat{b}+1}{s+1}\right).$$

From this, we can conclude that Algorithm 1 satisfies $(b, \varepsilon)$-resilience (Definition 3.1) with:

$$\varepsilon \in \mathcal{O}\left(\sqrt{\frac{1}{nT} + \frac{\hat{b}+1}{sT}} + \frac{\hat{b}+1}{s+1}G^2.\right)$$

$\square$

### A.8 PROOF OF LEMMA A.4

**Lemma A.4.** *Recall the event*

$$\Gamma = \left\{\forall t \leq T, \forall i \in \mathcal{H}, b_i^t \leq \hat{b}\right\}. \tag{6}$$

*Let $p < 1$ and assume $s$ and $\hat{b}$ are such that $b/n < \hat{b}/s+1 < 1/2$ and*

$$s \geq \min\left\{n-1, D\left(\frac{\hat{b}}{s}, \frac{b}{n-1}\right)^{-1} \ln\left(\frac{T|\mathcal{H}|}{1-p}\right)\right\}, \tag{7}$$

*with $D(\alpha, \beta) = \alpha \ln(\alpha/\beta) + (1-\alpha)\ln(1-\alpha/1-\beta)$.*

*Then the event $\Gamma$ holds with probability at least $p$.*

*Proof.* The proof is similar to the proof of Lemma 1 in Allouah et al. (2024a).

In our case, $b_i^t \sim \mathrm{HG}(n-1, b, s)$. Hence, by Lemma 13 in Allouah et al. (2024a), we have:

$$\mathbb{P}(b_i^t \geq \hat{b}) \leq \exp\left(-sD\left(\frac{\hat{b}}{s}, \frac{b}{n-1}\right)\right). \tag{14}$$

Assuming $s$ satisfies Equation (7), we have:

$$\mathbb{P}(b_i^t \geq \hat{b}) \leq \frac{1-p}{|\mathcal{H}|T}. \tag{15}$$

We can conclude by taking the union over all the events $\{b_i^t \geq \hat{b}\}$ for $i \in \mathcal{H}$ and $t \leq T$.

$\square$

## A.9 Proof of Lemma 4.1

**Lemma 4.1.** *If the number of samples $s$ satisfies :*

$$s \geq \left\lceil \max \left\{ \frac{1}{(1/2 - b/n)^2}, \frac{3}{b/n} \right\} \ln \left( \frac{4T|\mathcal{H}|}{1 - p} \right) \right\rceil + 2, \tag{3}$$

*then there exits $\hat{b}$ such that $\Gamma$ holds with probability at least $p$ and $\frac{\hat{b}}{s+1} \in \mathcal{O}\left(\frac{b}{n}\right)$*

*Proof.* The proof follows from the proofs of Lemma 2 and Lemma 4 in Allouah et al. (2024a), in which $T$ is replaced by $T|\mathcal{H}|$. □

## B Simulating the Effective Adversarial fraction

Algorithm 2 shows the pseudo-code of the procedure we use to choose $s$ and $\hat{b}$.

---

**Algorithm 2** Hyperparameters selection

---

**Require:** total number of nodes $n$, total number of attackers $b$, number of iterations $T$, grid of $s$ values $G_s$, number of simulations $m$, desired fraction threshold $\frac{b}{n} \leq q < \frac{1}{2}$, a hypergeometric law simulator HG.

1: $h = n - b$
2: **for** $s \in G_s$ **do**
3:     **for** $j = 1, \ldots, m$ **do**
4:         Draw $h \times T$ samples $b_i^t$ from $\mathrm{HG}(n - 1, b, s)$
5:         Compute $\hat{b}_s^{(j)} = \max_{i \in \mathcal{H}, t \in [1, \ldots, T]} (b_i^t)$
6:     **end for**
7:     Compute $\hat{b}_s = \max_{j = 1, \ldots, m} \hat{b}_s^{(j)}$
8:     Compute $\kappa_s = \frac{\hat{b}_s}{s+1}$
9: **end for**
10: **return** $(s, \hat{b}_s)$ such that $s$ is the smallest satisfying $\kappa_s \leq q$

---

**Remark 1.** This algorithm always returns a tuple $(s, \hat{b})$, provided the grid $G_s$ is large enough. For instance, if $n - 1 \in G_s$, we know that $\hat{b}_{n-1} = b$ and $\kappa_{n-1} = \frac{b}{n} \leq q$.

**Remark 2.** Notice that this algorithm does not take as input the probability parameter $p$. For large enough values of $m$, the algorithm returns a high probability upper bound $\hat{b}$ on the maximal number of encountered adversaries, without estimating this high probability. In practice, small values of $m = 5$ or $m = 10$ are enough for the upper bound to hold, especially for large values of $n$. This can be seen in Figure 3, where the confidence intervals are quite concentrated. An alternative, more precise method would be to compute the CDF of $\hat{b}_s$ for each value of $s$ and return the desired quantile corresponding to $p$. However, due to the absence of an easy closed-form of the CDF of a hypergeometric variable, this method also requires constructing an empirical CDF through simulations.

## C Experiments

### C.1 Experimental details

To run our experiments, we used a server with the following specifications:

- HPe DL380 Gen10
- 2 x Intel(R) Xeon(R) Platinum 8358P CPU @ 2.60GHz
- 128 GB of RAM
- 740 GB ssd disk

- 2 Nvidia A10 GPU cards

For the model architectures, we use the compact notation that can be found in El Mhamdi et al. (2021); Allouah et al. (2023):

L(#outputs) represents a fully-connected linear layer, R stands for ReLU activation, S stands for log-softmax, M stands for 2D-maxpool (kernel size 2), B stands for batch-normalization, and D represents dropout (with fixed probability 0.25). C(#channels) represents a fully-connected 2D-convolutional layer with (kernel size 3, padding 1, stride 1) for CIFAR-10 and (kernel size 5, padding 0, stride 1) for MNIST and FEMNIST.

| Dataset | MNIST | CIFAR-10 |
|---|---|---|
| Total number of nodes | $100, 30$ | 20 |
| Number of attackers | $10, 6$ | 3 |
| Heterogeneity | $\alpha = 1$ | $\alpha = 10$ |
| Model Type | CNN | CNN |
| Model Architecture | C(20)-R-M-C(20)-R-M-L(500)-R-L(10)-S | C(64)-R-B-C(64)-R-B-M-D-C(128)-R-B-C(128)-R-B-M-D-L(128)-R-D-L(10)-S |
| Weight L2 regularization | $10^{-4}$ | $10^{-2}$ |
| Learning Rate | 0.5 | 0.5 for $0 \leq t \leq 500$ 0.1 for $500 \leq t \leq 1000$, 0.02 for $1000 \leq t \leq 1500$, 0.004 for $1500 \leq t \leq 2000$ |
| Batch size | 25 | 50 |
| Momentum | 0.9 | 0.99 |
| Number of iterations | 200 | 2000 |
| Number of seeds | 2 | 3 |

Table 1: Experimental Details for MNIST and CIFAR-10

| Dataset | FEMNIST |
|---|---|
| Total number of nodes | 30 |
| Number of attackers | 3 |
| Heterogeneity | $\alpha = 10$ |
| Model Type | CNN |
| Model Architecture | C(64)-R-M-C(128)-R-M-L(1024)-R-L(62)-S |
| Weight L2 regularization | $10^{-4}$ |
| Learning Rate | 0.1 |
| Batch size | 50 |
| Momentum | 0.99 |
| Number of iterations | 500 |
| Number of seeds | 2 |

Table 2: Experimental Details for FEMNIST

For Figure 3, we fix $T = 200$, and we run $5$ simulations to build the confidence intervals for each point.

## C.2 COMPARISON TO BASELINES

We compare our method with the following baselines:

- **CS+** algorithm from Gaucher et al. (2025).
- The adaptive clipping version of **ClippedGossip** from He et al. (2022).
- **GTS**, the adaptation of **NNA** algorithm from Farhadkhani et al. (2023) to sparse graphs, as implemented by Gaucher et al. (2025).

We use ALIE attack, previously presented in the paper, and the Dissensus attack by He et al. (2022). The latter is more adapted to fixed graphs, as it leverages the structure of the gossip update to be made by the honest nodes.

For fixed $n$,$s$ and $b$, since our method uses $K = \frac{n \times s}{2}$ model exchanges, we generate a random connected graph with $K$ edges. To generate such a graph, we first generate a random spanning tree, using the Networkx implementation. Next, we add random edges to the graph until reaching $K$ edges.

*Remark* C.1. The graph construction here is more realistic than the one used in previous works (He et al., 2022; Gaucher et al., 2025), where the honest subgraph is generated first before the Byzantine connections are added. As a result, the honest subgraph may not satisfy the connectivity assumptions laid out by He et al. (2022) and Gaucher et al. (2025).

*Remark* C.2. The previous works on fixed-graph robust learning (He et al., 2022; Gaucher et al., 2025) assume that each honest node can have at most a certain number of Byzantine neighbors. This quantity is used as a parameter in the algorithm to choose the clipping threshold. Given that in our experiments, we generate random communication graphs, we fixed this quantity to be equal to $\hat{b}$. This only works if we assume the attackers' positions are random on the graph (which was the case in these experiments). In the case where the attackers' positions are also adversarial, it is necessary to choose this upper bound to be equal to $b$, the total number of Byzantine nodes, which tremendously reduces the performance, since the honest nodes are required to clip many more neighbors. It is important to note that this limitation is bypassed by our algorithm, which is also robust to the attackers changing their identities from one iteration to another.

**Analysis** The performance gain of RPEL with respect to the baselines is most impressive when the connectivity is low (bottom-left picture in Figure 4 and Figure 5). The worst client performance, which tracks the accuracy of the honest client who got the worst final accuracy, shows a clear and consistent advantage of using RPEL.

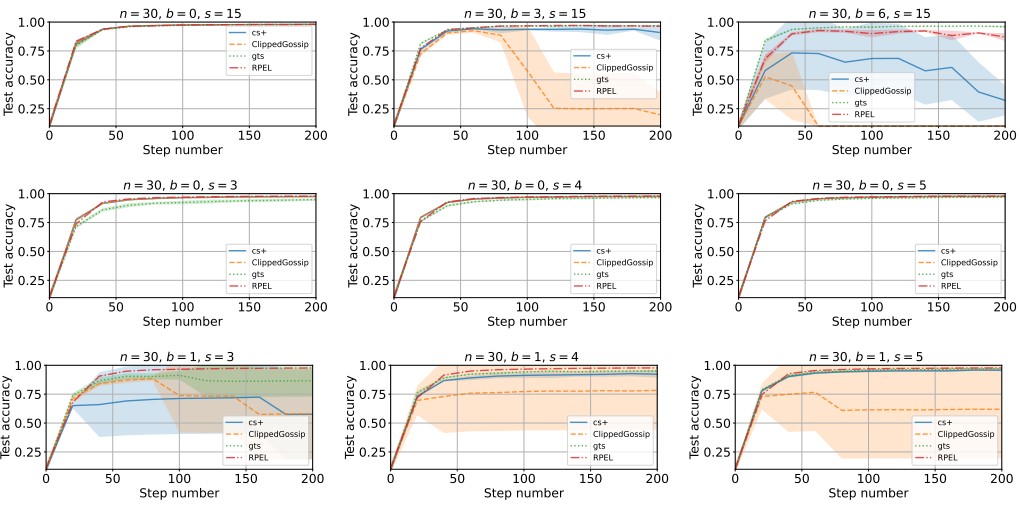

Figure 4: Average test accuracy on MNIST - ALIE attack

## C.3 OTHER EXPERIMENTS

In this section, we include a few additional experiments.

**Local steps** We conduct some experiments on CIFAR-10, for the same default setup, but with 3 local steps at each iteration. Figure 15, Figure 16, and Figure 17 show the test accuracies for $s = 6$, $s = 10$ and $s = 19$ respectively. Unsurprisingly, the performance is better than using a single local step before communication (Allouah et al., 2024a). The algorithm converges much faster to 75%+ accuracy. However, still in this scenario, using only 6 neighbors results in comparable performance with all-to-all communication.

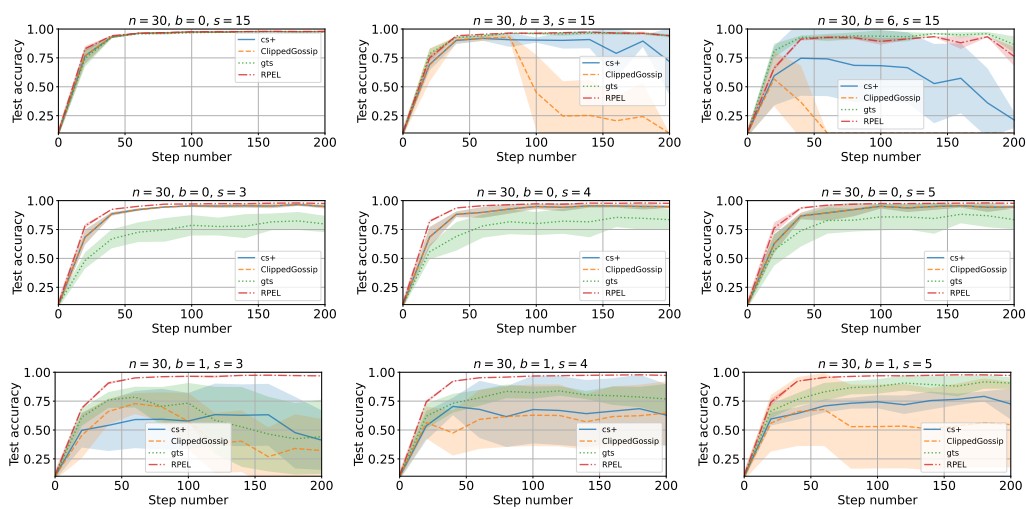

Figure 5: Worst test accuracy on MNIST - ALIE attack

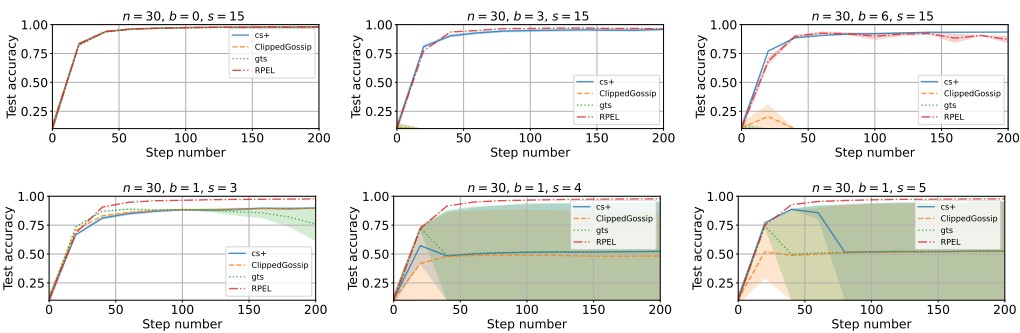

Figure 6: Average test accuracy on MNIST - Dissensus attack

**Higher heterogeneity.** We run a few other experiments with higher heterogeneity (lower values for $\alpha$) on the CIFAR-10 dataset. RPEL still performs well and remains robust to the different attacks.

**FEMNIST** Figure 20, Figure 21 show the convergence results for FEMNIST dataset with $n = 30$ and $b = 3$. For comparison, we provide the results in the absence of attackers in Figure 18 and Figure 19. The same conclusions about the good performance of RPEL can be drawn here. Implementation details for FEMNIST experiments are included in Table 2.

# D    DISCUSSION

We previously stated that the Push variant of Epidemic learning fails to Byzantine flooding attacks, since the honest nodes cannot control who sends them the updates. The Pull variant reduces the attack surface and gives back control to the honest nodes, which explains the scalability benefits of RPEL.

However, the Pull variant can be vulnerable to Denial of Service attacks, where the Byzantine nodes may intentionally delay sending their models to slow down the learning. In the synchronous model presented in the paper, this attack is not possible. In other words, if the honest nodes are guaranteed to respond promptly to model requests, the Byzantine nodes have no option but to do the same. Solving the issue for the general asynchronous model, where honest nodes can also be arbitrarily slow, is not straightforward and can be an interesting future research question.

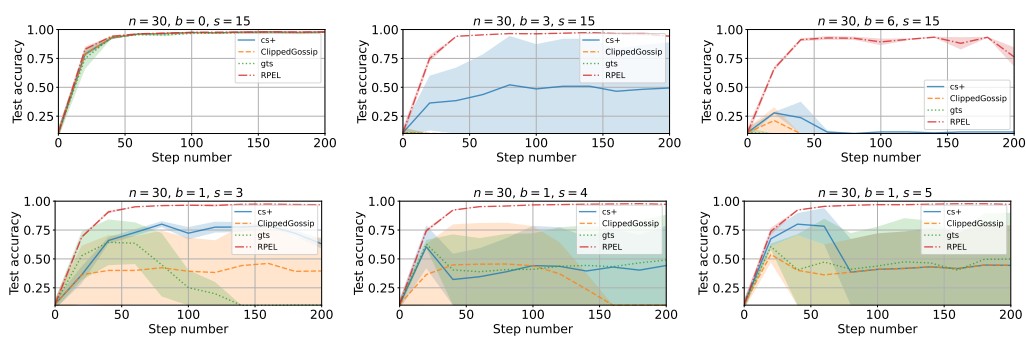

Figure 7: Worst test accuracy on MNIST - Dissensus attack

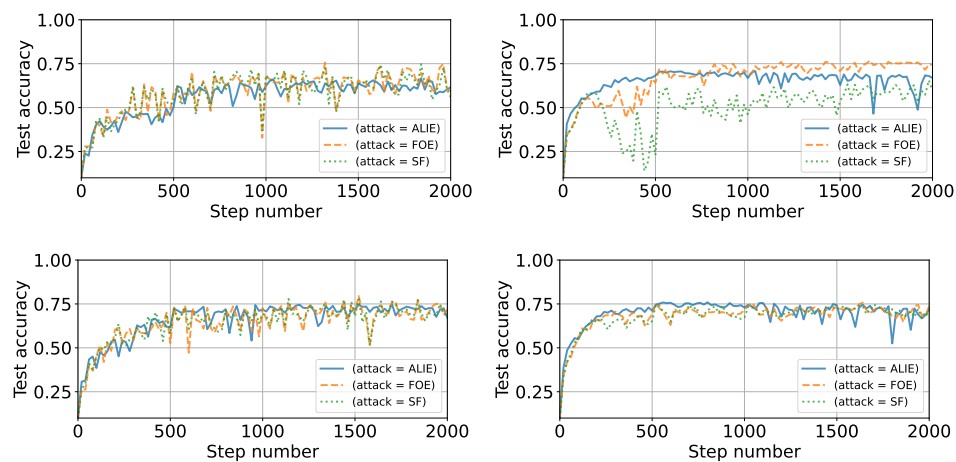

Figure 8: Test accuracies obtained on CIFAR-10 with $n = 20, b = 3$ (Left) $s = 6$, (Right) $s = 19$. (First row) $\alpha = 0.5$, (Second row) $\alpha = 1$

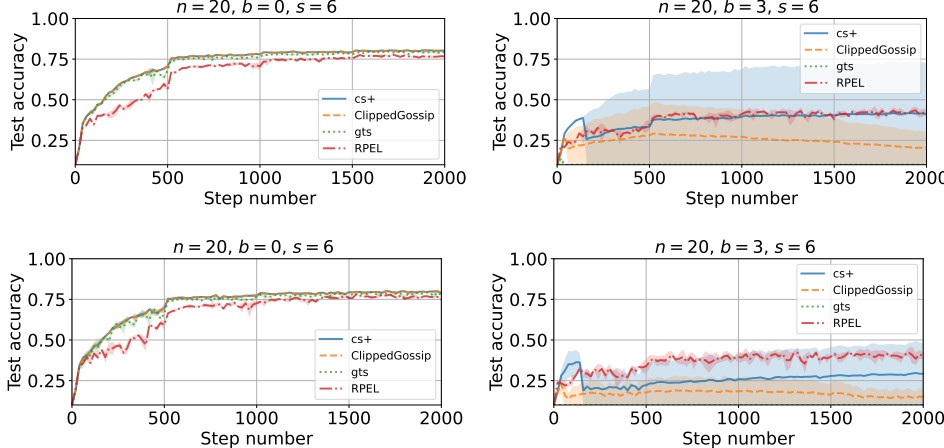

Figure 9: Test accuracies obtained on CIFAR-10 with Dissensus attack, $\alpha = 1$ and 1 local step. (Top row) Average performance. (Bottom row) Worst performance.

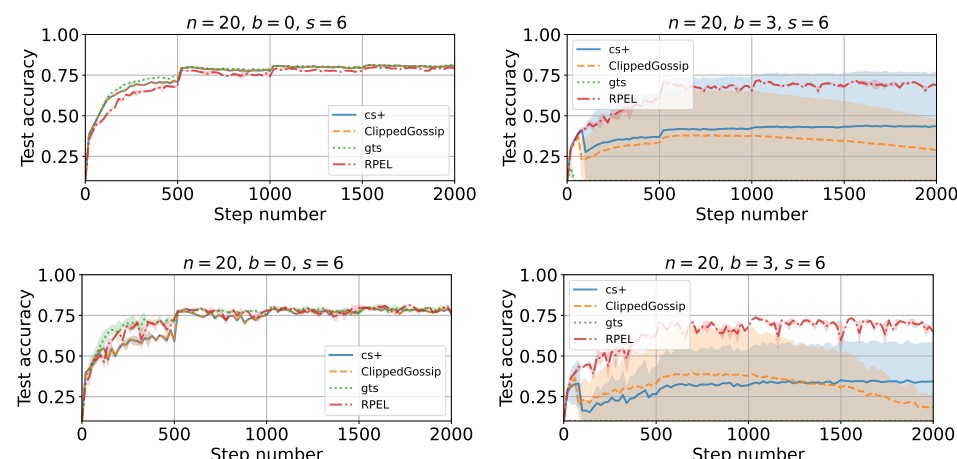

Figure 10: Test accuracies obtained on CIFAR-10 with Dissensus attack, $\alpha = 1$ and 3 local step. (Top row) Average performance. (Bottom row) Worst performance.

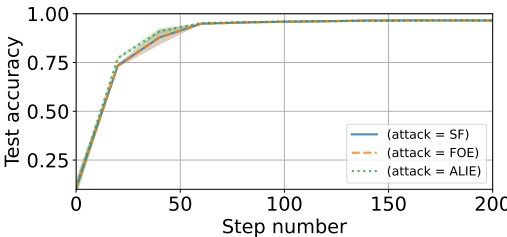

Figure 11: MNIST, $n = 100$, $f = 8$, $s = 15$

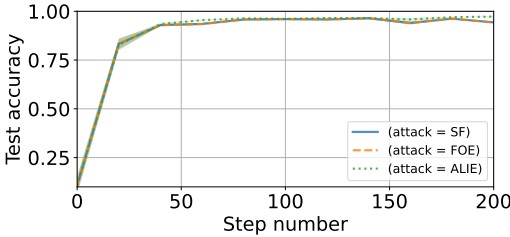

Figure 12: MNIST, $n = 30$, $f = 5$, $s = 15$

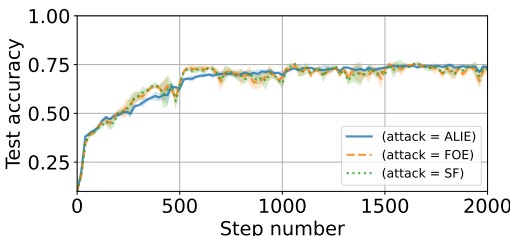

Figure 13: CIFAR-10, $n = 20$, $f = 2$, $s = 6$

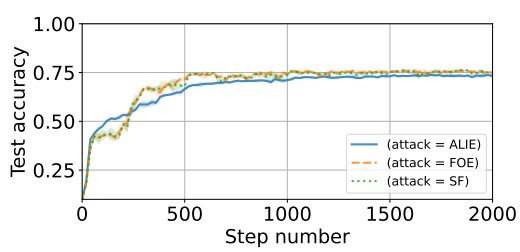

Figure 14: CIFAR-10, $n = 20$, $f = 2$, $s = 19$

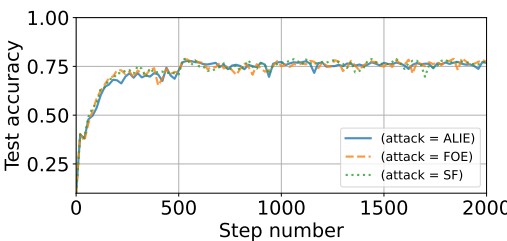

Figure 15: CIFAR-10, $n = 20$, $f = 3$, $s = 6$, 3 local steps

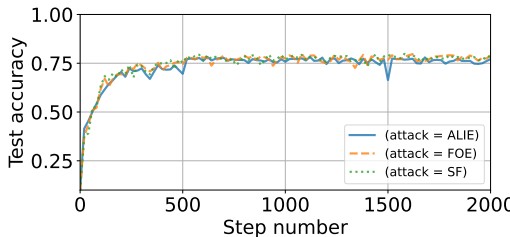

Figure 16: CIFAR-10, $n = 20$, $f = 3$, $s = 10$, 3 local steps

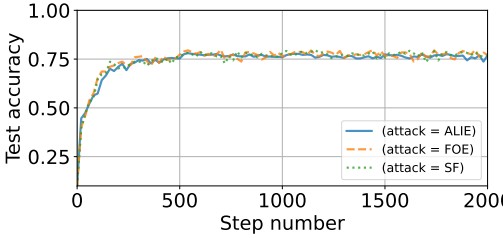

Figure 17: CIFAR-10, $n = 20$, $f = 3$, $s = 19$, 3 local steps

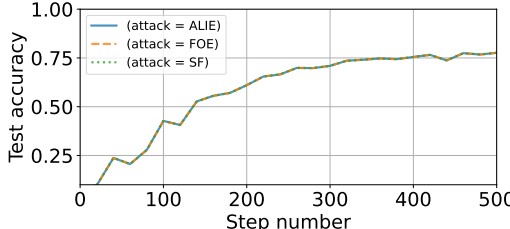

Figure 18: FEMNIST, $n = 30$, $f = 0$, $s = 6$

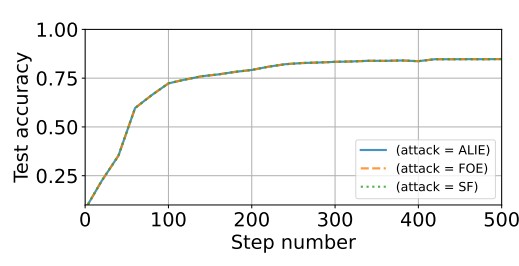

Figure 19: FEMNIST, $n = 30$, $f = 0$, $s = 6$, 3 local steps

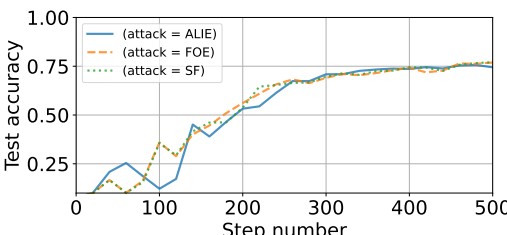

Figure 20: FEMNIST, $n = 30$, $f = 3$, $s = 6$

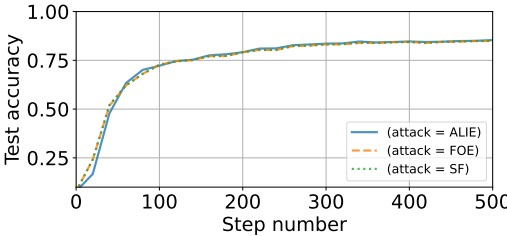

Figure 21: FEMNIST, $n = 30$, $f = 3$, $s = 6$, 3 local steps

