# OpenReview forum: "Robust and Efficient Collaborative Learning"
_ICLR.cc/2026/Conference — Submitted to ICLR 2026_

### Official Review · Reviewer_jECa · 2025-10-30

**Soundness:** 3
**Presentation:** 2
**Contribution:** 2
**Rating:** 4
**Confidence:** 3

**Summary:**

This paper proposes Robust Pull-based Epidemic Learning (RPEL), a decentralized Byzantine-robust collaborative learning algorithm.
Unlike traditional robust federated or decentralized learning methods that require all-to-all communication (O(n²) cost) or a trusted central server, RPEL uses a randomized pull-based epidemic protocol, where each node periodically pulls model parameters from a small, random subset of peers (O(log n) neighbors). Theoretical analysis shows convergence in nonconvex settings with data heterogeneity and omniscient adversaries. Empirical results on MNIST and CIFAR-10 show that RPEL achieves accuracy comparable to all-to-all robust methods but with significantly reduced communication.

**Strengths:**

- An interesting idea: Using pull-based epidemic communication for robust decentralized learning is neat. It meaningfully reduces communication overhead while preserving robustness.

- Solid theoretical grounding: The convergence proofs are rigorous and build upon established robust aggregation theory (e.g., Allouah et al. 2023). The notion of an Effective Adversarial Fraction provides an intuitive probabilistic understanding of robustness under random sampling.

- Empirical competitiveness: Despite reduced communication, RPEL maintains similar accuracy to all-to-all robust methods under several common Byzantine attacks (SF, FOE, ALIE).

**Weaknesses:**

- Limited empirical validation: Experiments are restricted to small datasets (MNIST, CIFAR-10) and small clusters (100 clients). The claims of large-scale scalability remain untested in actual large or high-dimensional systems.

- Attack model realism: Tested adversaries are restricted (sign-flipping, ALIE, FOE). Many powerful and classic attacks (Krum, Bulyan, Fang attack) are ignored, not to mention analysis against coordinated or sampling-aware adversaries that could exploit the pull mechanism.

- Parameter sensitivity: Robustness depends critically on the correct choice of sampling size s and effective adversary bound. These hyperparameters are tuned heuristically without sensitivity analysis.

**Questions:**

- How sensitive is RPEL to the sampling parameter s? Does small mis-specification significantly reduce robustness?

- Could adaptive adversaries bias the pull selection process or predict which nodes will be queried?

- How would RPEL perform when the network topology is inherently sparse and not fully connected?

- Can the concept of the Effective Adversarial Fraction be empirically validated beyond simulation (e.g., real-world distributed setting)?

---

> ### Author Response · Authors · 2025-11-19
>
> We thank the reviewer for their feedback, and we address the noted concerns in the rebuttal below.
>
> ## On the empirical validation.
>
> - To the best of our knowledge, works in decentralized learning usually remain way below the $100$-node setting we tested in our experiments, and hover mostly around $15$ or $20$ nodes. Going beyond $100$ nodes presents extreme infrastructure challenges that go beyond the purpose of this work. One such challenge stems from the fact that experiments in decentralized learning research rely on simulating the decentralized procedure on a single machine, with at most a few GPUs ($2$ in our case). Since each node needs to keep a distinct version of the model, this creates huge memory issues as $n$ becomes larger.
>
> - Following the reviewer's suggestion, we are currently running a few additional experiments with the additional attacks. However, we would like to underline that the attacks used in our experiments are quite powerful and the adversarial setting is quite strong. The adversaries are allowed to send different updates to different nodes in the same iteration. Our response to the second question addresses why adaptive attackers cannot exploit the pull mechanism.
>
>
> ## Response to the questions.
>
> - **On the sensitivity to sampling parameter.**
>
> From our experiments, as long as the event $\\Gamma$, defined in the paper, occurs, the experiments succeed (in the sense that the test accuracies are as expected, given the fraction of Byzantine attackers). When $s$ is chosen too small for $\\Gamma$ to occur, no guarantees can be provided. In that case, the experiments might still occasionally succeed in practice if a Byzantine majority is encountered only in the earlier training rounds.
> - **On adaptive attack strategies.**
>
> Contrary to the push mechanism, the pull selection is robust to adaptive strategies, as the honest nodes always select uniformly at random. Our attack model assumes the attackers control a certain number of nodes $b$, but the remaining nodes remain intact. Additionally, we assume that each participant knows the list of all the other participants, which excludes any adaptive strategy at the selection step.
> - **On inherently sparse topologies.**
>
> The RPEL algorithm can be easily extended to the inherently sparse communication topologies, by making each node select a uniform sample of $s$ nodes from its neighborhood. From a theoretical point of view, it is not a trivial question to show under which circumstances this version of RPEL would outperform the other baselines in this case. However, intuitively, for the same reason why RPEL outperforms all-to-all robust methods in the fully connected case from a communication point of view, by eliminating the need to use all the available links all the time, we expect a similar communication reduction benefit in the sparse case.
> - **Effective Adversarial fraction beyond simulations.**
>
> The concept is quite practical, and we do not see any reason why it would not extend to real-word distributed settings. Knowing the total number of participants and an accurate estimation of an upper bound on the number of Byzantine participants, and fixing the number of selected neighbors $s$, each node can estimate the Effective Adversarial fraction as a probabilistic worst case of the fraction of adversaries it is exposed to throughout the training procedure.

---

### Official Review · Reviewer_Lskz · 2025-10-31

**Soundness:** 2
**Presentation:** 2
**Contribution:** 2
**Rating:** 4
**Confidence:** 4

**Summary:**

This paper introduces the Robust Pull-based Epidemic Learning (RPEL) protocol, a novel and scalable collaborative approach designed to ensure robust learning despite omniscient Byzantine adversaries. By employing a pull-based epidemic communication strategy, RPEL significantly reduces the communication complexity. The work establishes rigorous finite-time convergence guarantees under non-convex settings and introduces the concept of the Effective Adversarial Fraction as a key robustness metric. Empirical results validate effectiveness of the proposed method against state-of-the-art attacks.

However, its practical impact is currently limited by two key areas needing improvement. First, the strict reliance on a synchronous communication model is a major drawback, so it’s necessary to address this by relaxing the assumption and evaluating performance under asynchronous conditions. Second, the baseline selection lacks clear justification, and the fairness of the comparison is questionable, given that baselines operated under sub-optimal conditions.

**Strengths:**

1. The paper proposes RPEL, representing the first scheme in a decentralized, serverless setting to simultaneously achieve communication efficiency and robustness against Byzantine attacks. This innovation in mitigating high communication costs is paramount for large-scale decentralized learning.
2. The theoretical analysis is highly rigorous, providing convergence guarantees under a general non-convex and data heterogeneous setting. Notably, it proves robustness against omniscient attacks, a threat model that is significantly more powerful than those typically assumed.

**Weaknesses:**

--The experimental evaluation lacks a clear justification for its specific choice of baselines. While the paper compares RPEL against CS+, ClippedGossip, and GTS in Appendix C.2, it fails to explicitly state the rationale for selecting these methods over others. Furthermore, the other relevant methods discussed in the related work, such as Remove-Then-Clip (RTC), were notably omitted from the comparison without any explanation, which leaves the comparative analysis feeling incomplete. It’s also questionable whether the comparison is fair, given that the baseline methods were operating under sub-optimal conditions.

--The paper should include more discussion on RPEL under asynchronous settings.. The entire theoretical framework, including the convergence and robustness guarantees, presumes that all participating nodes possess a global clock and respond promptly to model requests. This assumption is often untenable in large-scale, geographically dispersed decentralized systems, where asynchronous settings are virtually a necessity. The authors concede this limitation in Appendix D, noting that while the proposed pull-based approach effectively thwarts Byzantine flooding attacks, it remains vulnerable to DoS attacks in a more general asynchronous model. Therefore, the paper should endeavor to relax this stringent assumption and includes further experiments and discussion on the method's performance and robustness under more realistic asynchronous conditions.

**Questions:**

Please see weaknesses.

---

> ### Author Response · Authors · 2025-11-19
>
> We appreciate the reviewer’s feedback and outline our replies to the raised issues in the rebuttal below.
>
> **On the experimental setting and baseline comparisons**
>
> We restricted our comparisons to the state-of-the-art robust decentralized methods, and we are currently running some experiments with the Remove-Then-Clip (RTC) method. However, we would like to underline that the purpose of (parts of) our experiments is to highlight the key limitations of fixed-graph methods, which require a high network connectivity and communication cost to ensure robustness. Hence, for a fair comparison, we fixed the communication and computation budget and optimized the hyperparameters of the baselines to achieve the best possible performance for each setting. We encourage the reviewer to explain what they meant by sub-optimal conditions.
>
> **On the asynchronous setting**
>
> Asynchronous learning with Byzantine behaviors is a highly interesting research question. We chose to restrict our work to the synchronized setting simply to isolate the question of communication efficiency. Indeed, the more realistic asynchronous setting broadens the attack surface, allowing DoS attacks as we acknowledge in the paper. That being said, solutions that extend robust methods from the synchronous setting to the asynchronous one usually only incur a linear overhead (e.g. Farhadkhani et. al. (2023)). In other words, to ensure robustness to DoS attacks, the usual recipe would be to sample $2s$ nodes instead of $s$ nodes, which still preserves the $\mathcal{O}(n \log n)$ communication complexity of RPEL. Nevertheless, extending our current theoretical analysis to this asynchronous variant is non-trivial and remains an open question for future work.

---

### Official Review · Reviewer_PQyh · 2025-10-31

**Soundness:** 1
**Presentation:** 3
**Contribution:** 1
**Rating:** 2
**Confidence:** 4

**Summary:**

This paper introduces a distributed collaborative learning framework called Robust Pull-based Epidemic Learning (RPEL). Instead of exchanging model parameters with all nodes in the network, RPEL adopts a pull-based strategy, where each node retrieves updates from only a small, randomly selected subset of peers. This design dramatically reduces communication overhead while maintaining learning performance. The key contribution lies in the theoretical analysis showing that such random sampling does not compromise robustness with high probability and still guarantees convergence of the overall learning process. The approach is notable because, despite its simplicity, it provides strong theoretical assurances for a randomized communication protocol: in every round, each honest node pulls updates from a random subset of peers. However, while the expected fraction of corrupted neighbors remains the same under random sampling, the actual fraction in any given round can fluctuate significantly, potentially deviating the training from benign behavior. Therefore, establishing worst-case bounds on the effective adversarial fraction is critical to ensuring the robustness of the proposed algorithm. The paper attempts to show these worst-case bounds.

**Strengths:**

1. It discusses a reasonable problem where there is a need for scalable and robust learning algorithms.
2. The paper attempts to rigorously analyze the algorithms and provides bounds for their claims.
3. The paper is very well written and easy to follow.

**Weaknesses:**

**Major Weaknesses**
1. The theoretical claims are not carefully analyzed which becomes an extreme downside as the main contribution of the paper is the theoretical guarantees of the proposed algorithm.

(a) Consider Lemma 4.1, it provides a lower bound for $s$ such that $\frac{\hat{b}}{s + 1} \in O(\frac{b}{n})$ with at least probability $p$. This essentially gives the lower bound on $s$ to limit the effective adversarial fraction by $O(\frac{b}{n})$ for every honest node throughout the training. This lemma becomes trivial if the lower bound is more than $n$ and does not provide meaningful guarantee.

For parameters in experimental results in Fig. 1, $T = 200, n = 100, b = 10, \mathcal{H} = 90$, if we consider the probability of failure $1-p = 0.2$, the actual lower bound on $s$ comes out to be:
$$s \geq 386,$$
It results in similar lower bounds for other experimental settings as well.

(b) Similarly, for eq. $7$ in Lemma A.4, the actual lower bound for $s$ is far more than used in the experimental settings for any reasonable probability of failure. As mentioned section 6.1, Eq 7 requires $s$ to be very large and hence $s$ is chosen through simulation which reflects that performed experiments don't enjoy the theoretical guarantees presented in earlier sections.

2. The notion of robustness in Definition 5.1 is not sufficient. The upper bound on bias $\Sigma ||v_i - \bar{v}_{\mathcal{U}}||^2$ may seem small and constant but it is proportional to $d$ (dimension of vectors). This can still result in very high bias in applications like collaborative learning where the model size is reasonably large.

The problem of robust mean estimation is very well studied at this point. The optimal bias guarantees are independent of the dimension proposed in [1]. Robust Aggregators like coordinate-wise median and geometric mean fail to give bias independent and are shown to provide robustness even in the centralized federated learning setup [2]. Even the theoretically optimal aggregators proposed in [1] are prone to attacks due to their computational bottleneck in centralized collaborative learning [3].

This highlights that aggregators and attacks used in this paper are not state-of-the-art. There has been a significant advancement in the literature of robust mean estimation in last 5-6 years which has been neglected in the analysis.

[1] I. Diakonikolas, G. Kamath, D. Kane, J. Li, A. Moitra, and A. Stewart, “Robust estimators in high-dimensions without the computational intractability,” in SIAM Journal on Computing, 2019.

[2] B. Zhu, L. Wang, Q. Pang, S. Wang, J. Jiao, D. Song, and M. I. Jordan, “Byzantine-robust federated learning with optimal statistical rates,” in AISTATS, 2023.

[3] Choudhary, S., Kolluri, A. and Saxena, P., 2024, May. "Attacking byzantine robust aggregation in high dimensions." in IEEE Symposium on Security and Privacy (SP) 2024.

**Questions:**

1.  Can authors provide the exact values of their effective adversarial fraction along with the probability of failure for the experimental results (using their theoretical results) ?
2. Why is there a uniform sampling in line 12-14 of Algorithm 1?

---

> ### Author Response · Authors · 2025-11-19
>
> We thank the reviewer for their review, which raises some important questions about our paper. We would nonetheless like to clarify why we respectfully disagree with several aspects of the assessment.
>
> **About the major weakness**
>
> The reviewer is correct in pointing out that the lemma can sometimes provide a trivial condition on the number of selected neighbors $s$ (when the right-hand-side is larger than $n$). This is due to the **constant and logarithmic factors** being potentially significant in the small $n$ regimes (e.g. the example given by the reviewer, as well as most of our experiments). However, at scale, assuming the proportion of Byzantine attackers is fixed, this condition provides values of $s$ much lower than $n$. For instance, fixing the Byzantine fraction to $10\\%$ (similar to the example provided by the Reviewer), but taking $n=10000$, the condition becomes $s\ge 524$ which is quite small (corresponding to only a $5\\%$ selection rate for each client).
>
> This theoretical condition is **sufficient** but by no means necessary, as the experiments demonstrate. Contrary to the reviewer's statement, "which reflects that performed experiments don't enjoy the theoretical guarantees presented in earlier sections", we argue that the experiments actually exhibit a stronger performance, since the required $s$ in practice is much smaller than the one given by the lemma.
>
> We also wish to emphasize that this limitation is clearly acknowledged in the paper, and in fact motivates the practical selection procedure (Algorithm 2) which selects $s$ through inexpensive simulations.
>
> As stated in the paper, the main purpose of this lemma is to show that, at scale, only a logarithmic portion of the participating nodes is required to obtain an Effective Byzantine fraction comparable (in Big O notation) to the standard Byzantine fraction, without the burden of all-to-all communication.
>
> We further note that our theoretical results require only the condition in Equation (7), which is substantially less restrictive than Equation (3) referenced by the Reviewer. Equation (3) is needed only for Corollary 5.7, which is an asymptotic result.
>
> **About robustness in High dimensions**
>
> We thank the reviewer for their remark on high dimensional robustness. We are familiar with the literature on the topic, but we view this question as orthogonal to the main focus of the paper, namely the dependence on the graph connectivity. Extending the high-dimensional procedures to our setting does not seem unfeasible and represents a promising direction for future work.
>
> **We now provide answers to the specific questions**
>
> 1. As discussed above, the theoretical condition on the choice of $s$ is quite loose in practice, especially at smaller scales. For our experiments, which remain way below the $10000$-node setting in the earlier example, the bound is not informative. However, it is straightforward to verify that the condition of Equation (7) in the appendix holds. This condition is sufficient for the event $\\Gamma$ to occur, though it does not guarantee that the Effective fraction $\frac{\hat{b}}{s+1}$ matches the order of the initial fraction $\frac{b}{n}$.
>
> 2. The uniform sampling procedure in the algorithm is mostly a theoretical device commonly used in non-convex optimization, due to the guarantees being on the average loss on the first $T$ iterations (c.f. Theorem 5.6). Examples of this approach appear in [1,2,3].
>
>
> We hope that these clarifications address the reviewer's concerns and respectfully ask them to reconsider their evaluation in light of the above.
>
> [1] Levy, K. Y., Kavis A., Cevher V. (2021) STORM+: Fully Adaptive SGD with Momentum for Nonconvex Optimization
>
> [2] Reddi, S. J., Hefny, A., Sra, S., Póczos, B., Smola, A. (2016). Stochastic Variance Reduction for Nonconvex Optimization.
>
> [3] Allen-Zhu, Z., Hazan, E. (2016). Variance Reduction for Faster Non-Convex Optimization.

---

> ### Comment · Reviewer_PQyh · 2025-11-21
>
> I thank the authors for their additional explanation.
>
> > This theoretical condition is sufficient but by no means necessary
>
> To my understanding, both eq (3) and eq (7) provide a lower bound to $s$ (which seems to be tight) for the effective adversarial fraction to be bounded with some probability. It implies that it is necessary for the algorithm to sample that many nodes to claim the limit on effective adversarial fraction. Now, in a simulation, even with less number of sampled nodes the effective adversarial fraction may remain bounded but that is just an empirical evidence. It not necessarily implies that these simulations will always exhibit strong performance. Wasn't this the whole point of giving lower bound on $s$ ?
>
> I agree for very large $n$, the lower bound will actually be useful but I would say such a large $n$ is not practical for distributed learning. It is good to show that number of sample required to maintain the order of corruption fraction scales with log, as shown in Allouah et al. (2024a) as well. Practical relevance of the bound is still questionable.
>
> Regarding the robustness in high dimensions, even though the authors are aware of these literature, the manuscript itself doesn't reflect it. Also, what is the reason to not to choose a correct notion of robustness as per the works mentioned earlier ? The main goal of the paper is to propose a robust decentralized learning algorithm which requires choosing a correct notion of robustness. The current definition of robustness (Def 5.1) doesn't provide any reasonable bias bound for high dimension (including the models considered in experiments). Even if you can limit the effective adversarial fraction, that much adversarial fraction is enough for targeted or untargeted attacks, for the aggregator used currently (shown in [2, 3] from previous comment).
>
> I agree with authors that they can extend their random sampling to those aggregators as well but that should the part of current work itself. At the very least, authors should consider using the correct definition of robustness from the state-of-the-art literature.

---

> > ### Author Response · Authors · 2025-11-23
> >
> > We appreciate the reviewer's detailed follow-up.
> >
> > **Additional clarifications regarding Equations (3) and (7).**
> >
> > - Equation (7) is sufficient for the event $\Gamma$ to occur with probability at least $p$. Recall that event $\Gamma$ occurs when no Byzantine majority is encountered at any iteration by any honest node. For the previous example given by the reviewer, i.e. $T=200, n=100, b=10$, when choosing $s=15$ and $b=7$ like we did in the experiments, it is easy to verify that Equation (7) holds (even with $p=0.99999$). To a certain extent, this shows that the bound provided by Equation (7) is quite reasonable and practical.
> >
> > - The bound provided by Equation (3) does require a high number of nodes to be meaningful, but it is fundamentally a theoretical, asymptotic result, as it guarantees that at the same time the event $\Gamma$ occurs, and the Effective Byzantine fraction is of the same order as the initial Byzantine fraction, in other words, $\frac{\hat{b}}{s+1} \in \mathcal{O}(\frac{b}{n})$. The latter is indeed meaningless unless $n$ is large. We highlighted this equation in the paper mainly because it facilitates the theoretical comparison with previously established lower bounds (Section 5.3). Indeed, to compare with the convergence bounds previously established in the literature, it is essential to establish the link between $\frac{b}{n}$ and $\frac{\hat{b}}{s+1}$.
> >
> > We hope this clarifies this point, and we are happy to address any further concerns the reviewer might have about our theoretical contribution.
> >
> > **Additional comment about robustness in high dimensions**
> >
> > We fully agree with the reviewer about the importance of treating the high dimensional case. However, to the best of our knowledge, no work has extended these new notions of robustness from the centralized/federated to the peer-to-peer setting, which is the focus of this work.
> >
> > We reiterate that our goal here is to isolate the scalability with respect to the number of participants. To do so in a clean way, we adopted a notion of robustness that is widely used in the recent robust decentralized learning literature (e.g. Gaucher et. al (2025)). We will add a discussion on the importance of scalability with respect to the dimension, which remain an open, non-trivial, question and an interesting avenue for future research.

---

### Official Review · Reviewer_ts2U · 2025-10-31

**Soundness:** 3
**Presentation:** 4
**Contribution:** 3
**Rating:** 4
**Confidence:** 3

**Summary:**

This paper proposes a way to make distributed, collaborative learning, robust to adversarial/byzantine attacks. The idea is that each agent/node can pull updates from a subset s of its neighbors. Among these, there are going to be some expected number of adversaries, but provided there are enough honest nodes in the neighborhood (something captured by a parameter defined as the "effective adversarial threshold") learning can remain robust to adversaries. The paper proposes a randomized algorithm called RPEL for client selection and aggregation, which comes with theoretical guarantees on single iteration expected error reduction and convergence guarantees, and its performance is validated empirically (particularly against all-communication byzantine robust baselines).

**Strengths:**

The paper is very well presented and motivated. It studies an important and challenging problem.

The algorithm comes with both theoretical guarantees and empirical support.

The presented algorithm requires less restrictive bounds on the communication overhead and lower bounds on connectivity with honest nodes than existing methods.

**Weaknesses:**

Not necessarily weaknesses but clarifications about some points might be helpful:

1. I understand they are conceptually different, and agree with the description of why it is important to do pulls (not pushes, which would allow for flooding by adversaries) in this paper’s setting. But how about technically/theoretically? Meaning, can you elaborate more, from a technical perspective, why pull-based and push-based lead to differing analysis?

2. Similarly, I did a brief review of the proofs, and I was finding references to similar proofs in the all-to-all communications setting of Farhadkhani et al., or to the work of Allouah et al. 2024a on robust learning. What are the key technical contributions of this paper? In other words, how does the setup of this paper add to the analytical complexity of establishing error bounds or convergence guarantees? This was not clear to me throughout, and having some clarification of the technical contributions might help.

3. Equation (7) seems to be key for Theorem 5.6., but it only appears in the appendix, which affected the flow of the paper, and including the condition in the main paper would be helpful.

4. While not crucial/important, I was curious about the code, and the anonymized link was appreciated, but other than the readme file, I encountered the error “The requested file is not found.” for other files.

5. Minor edit suggestion: I believe the sentence in 248-249 should read the other way around. Meaning: "We call the ratio \frac{b}{s+1} the Effective adversarial fraction."

**Questions:**

While this paper was overall clear and interesting to me, I would appreciate clarification on the following two points during the rebuttal period, to help better inform my final assessment.

1. What are the technical differences between push-based and pull-based epidemics? (more details above in the weakness section).

2. What are the technical contributions/differences of this paper's theoretical results on convergence or per-round error reduction compared to existing works such as Farhadkhani et al. and Allouah et al., which are referenced often in the appendix, too? (more details above in the weakness section).

---

> ### Author Response · Authors · 2025-11-19
>
> We thank the reviewer for his/her feedback. We provide the clarifications requested by commenting on the points mentioned in the Weaknesses section.
>
> 1. Besides the possible adversarial flooding described by the Reviewer, both the algorithm and the analysis of push-based epidemic learning do not extend nicely to support Byzantine participants. From a conceptual perspective, honest nodes would receive a distinct number of updates at each iteration, and would be forced to use different aggregation rules (robust aggregation rules usually take as parameters the number of honest nodes and an upper bound on the number of Byzantine nodes, and both of these would be different across the honest clients). From a technical perspective, first the flooding issue entails that the logarithmic benefits cannot be realized. Second, assuming the Byzantine nodes respect the push protocol, it is not clear how to define a notion similar to the Effective Byzantine fraction. The analysis is further complicated by the fact that each aggregation made by client $i$ at iteration $t$ incurs an error involving a different $\kappa_i^t$.
>
> 2. The proofs included in the appendix represent our original contribution. Some additional proofs, which we reference in the appendix are used to complement our analysis. For instance, a key technical contribution of our work is proving the ($\alpha,\lambda)$-reduction result, summarized in Lemma 5.2. This Lemma controls the single iteration error variation.  We subsequently used the results from Farhadkhani et al. (2023) to conclude the convergence after $T$ iterations. We slightly adapted the choice of $\eta$ in our setting to get better convergence bounds, and we only included the adaptation step rather than the whole proof which can be found in Farhadkhani et al. (2023). (Notice for instance that our choice of $\eta$ results in a better dependence on ($\sigma^2+G^2$)).
>
> 3. Our choice to put Equation (7) in the appendix is merely due to the space limitation. We understand that is affects the flow of the paper and we will include it in the main paper for the final version.
>
> 4. We thank the reviewer for pointing out the issue with the anonymized link. On our end, it seems that all the files are available, but we will investigate this issue.
>
> 5. We thank the reviewer for the minor edit suggestion, which we will take into account in the revised version.

---

### Meta-Review · Area_Chair_4Z92 · 2025-12-22

**Summary:**

The paper introduces RPEL, a new method for robust collaborative learning that does not require a central server and scales favorably with the number of nodes. Overall, the reviewers appreciated the approach and found the writing and exposition clear.

However, the paper received negative scores only. The major concerns were related to lack of clarity regarding the technical contribution (Reviewer ts2U), applicability to high-dimensional settings (Reviewer PQyh), applicability under asynchronous updates (Reviewer Lskz) and simplicity of the experimental setup (Reviewer jECa).

Overall, the paper shows promise, but may benefit from extending its theoretical or empirical analysis to other/more general settings.

**Reviewer Concerns:**

The authors responded to all reviewers. Unfortunately, only reviewer PQyh responded before discussions were closed, reiterating their concerns regarding the validity of the results in high-dimensional settings. Overall, the authors argued that issues of high-dimensionality were orthogonal to the present submission, while reviewer PQyh regarded such settings as important.

Regarding the remaining concerns:

- the authors responded to the questions of reviewer ts2U
- they argued that concerns about asynchronous updates are orthogonal to this work
- they argued that experimental are similar to prior work and promised to provide more results (however, I was not able to find such updates)

**Reviewer Scores:**

Overall, it is plausible that reviewer ts2U could have raised their score, as the authors did clarify the technical contributions. However, it is unlikely that the other reviewers would have raised scores substantially. This is because the work does not currently extend to the settings they requested.

---

### Decision · Program_Chairs · 2026-01-26

Reject